



# Stochastic Modelling of Thermokarst Lakes: Size Distributions and Dynamic Regimes

Constanze Reinken[1], Victor Brovkin[1], Philipp de Vrese[1], Ingmar Nitze[2], Helena Bergstedt[3], and Guido Grosse[2,4]

[1]Max Planck Institute for Meteorology, Bundesstraße 53, 20146 Hamburg, Germany
[2]Alfred Wegener Institute Helmholtz Centre for Polar and Marine Research, Telegrafenberg A45, 14473 Potsdam, Germany
[3]b.geos GmbH, Industriestrasse 1, Korneuburg, 2100, Niederösterreich, Austria
[4]University of Potsdam, Institute of Geosciences, Karl-Liebknecht-Str. 24-25, 14476 Potsdam, Germany

**Correspondence:** Constanze Reinken (constanze.reinken@mpimet.mpg.de)

**Abstract.**

Thermokarst lakes are among the most common and dynamic landscape features in ice-rich permafrost lowland regions. They influence carbon, water and energy fluxes between atmosphere and land surface and are an important component of Arctic lowland hydrology. Despite their significant role in the climate system, thermokarst lakes are only rudimentarily or
not at all represented in Earth system models (ESMs). Attempts at stand-alone modelling of their dynamics have mostly been limited to the scale of individual lakes. Because lake formation, expansion, and drainage depend on small-scale surface and sub-surface heterogeneities that are difficult to measure, a deterministic modelling-approach would be a challenge at the regional or pan-Arctic scale. We therefore treat these processes as probabilistic across a landscape and create a model of thermokarst lake dynamics using stochastic approaches. With the inclusion of stochasticity and volatility, our method allows us to account for
the diversity of individual lake behaviour that results from the small-scale differences in environmental conditions. We present idealized simulations and, additionally, test novel high-resolution remote sensing data products that capture annual lake areas for model initialization and the calibration of inherent or climate-induced lake dynamics. Our model is able to capture three plausible regimes by incorporating the main processes behind thermokarst lake dynamics and represents a new step towards stochastic representation of permafrost landscapes in ESMs. Furthermore, our findings emphasize the importance of continued
remote sensing data retrieval and additional data products containing information on past thermokarst lake behaviour for model parameterization.

## 1 Introduction

It is estimated that permafrost underlies about 13-18% of the northern hemisphere's unglaciated surface (Zhang et al., 2000),
and that roughly $1300 \pm 200$ gigatons of carbon are stored across all permafrost regions (Hugelius et al., 2014). This carbon can



become subject to increased biodegradation upon warming and subsequent permafrost thaw, and released into the atmosphere, thereby increasing the greenhouse effect. This so-called permafrost carbon feedback is especially relevant, considering that the Arctic is warming up to four times faster than the rest of the globe. However, it is either inadequately or not at all represented in current Earth System Models (ESMs), as they miss essential processes and feedbacks of permafrost landscapes, such as abrupt

thaw (Schädel et al., 2024; Matthes et al., 2025; Brovkin et al., 2025). Using inventory models, (Turetsky et al., 2020) found that carbon emissions through abrupt thaw could increase permafrost carbon emissions by 40%, which means that current ESMs are likely underestimating the climate effects of degrading permafrost.

A main abrupt thaw process is so-called thermokarst, which is the process of landscape changes due to melting of ground ice and resulting ground subsidence. An estimated 20-40% of the northern permafrost domain is covered by thermokarst-affected

landscapes (Olefeldt et al., 2016; in 't Zandt et al., 2020). One prominent product of these landscape changes are thermokarst lakes, that form when water pools in thermokarst depressions. The earliest of these lakes can be traced back to the Late Glacial transition time at the end of the Late Pleistocene and the early Holocene between circa 14 to 10 ka BP, during which there was a strong warming and wetting trend in the Arctic (Mann et al., 2002; Velichko et al., 2002; Kaufman, 2004; Grosse et al., 2013; Brosius et al., 2021). A synthesis of lake initiation records found two peaks in lake formation rates at 13.2 and 10.4 ka

BP (Brosius et al., 2021). The process of depression and lake formation has continued to occur since then and current global warming trends could lead to an acceleration of thermokarst lake dynamics. Due to thermal erosion along their shorelines, thermokarst lakes expand radially over time. Accordingly, their shapes are often highly circular. However, due to interactions with ground ice distribution and preferential wind patterns as a main driver for waves and currents (Côté and Burn, 2002), there are also regions with oriented thermokarst lakes that have been described as elliptical, triangular, rectangular and egg-, clam-

or D-shaped (Grosse et al., 2013; Hinkel et al., 2005; Morgenstern et al., 2011).

Thermokarst lakes can also grow vertically (Fedorov et al., 2014; in 't Zandt et al., 2020). Because of the lower albedo, higher absorption of long-wave radiation, and higher heat storage capacity of water in comparison to ice or dry ground, they often accelerate the thawing of surrounding permafrost, including potential melting of additional ground ice (Hopkins, 1949; Mackay, 1992; Plug and West, 2009; Jorgenson et al., 2010; Grosse et al., 2013; in 't Zandt et al., 2020). If they are deep

enough not to freeze to the bottom in winter, they are typically underlain by a so-called talik, which is a layer of year-round unfrozen ground (Lachenbruch et al., 1962; Mackay, 1992; West and Plug, 2008; in 't Zandt et al., 2020). The number of these floating-ice lakes have increased with climate change, which means that the effect on permafrost thaw will likely become stronger in the future (Arp et al., 2018, 2015; Surdu et al., 2014).

Thermokarst lakes can drain gradually or abruptly in a matter of only a few days due to a variety of triggers and mechanisms

that depend on climatic and environmental conditions. Drainage can generally be categorized into lateral and internal drainage, with internal drainage happening through sub-surface drainage networks once the lake talik, penetrates the permafrost layer (Hopkins, 1949; Mackay, 1988) and therefore being more common in discontinuous permafrost (Chen et al., 2023). A growing lake can increase its likelihood of internal drainage by creating sub-surface drainage channels through facilitating permafrost degradation (Yoshikawa and Hinzman, 2003; Rowland et al., 2011). Lateral drainage can occur when a lake expands and

either a topographic drainage gradient is reached, a newly formed channel network from degrading polygonal ice wedges or





thermo-erosion allows seepage, or a positive water balance leads to bank overflow (Mackay, 1988; Hinkel et al., 2007). New surface drainage channels can also be created by external factors, such as coastal erosion and tapping by a river, stream or other lake (Mackay, 1988; Hinkel et al., 2007; Arp et al., 2010; Jones et al., 2020). High lake drainage rates have also been linked to a combination of abundant rain- and snowfall and extremely warm mean annual air temperatures (Nitze et al., 2020).

Despite these numerous drainage mechanisms, some lakes have persisted for several thousands of years (Edwards et al., 2016; Bouchard et al., 2017; Anderson et al., 2019) and in some cases, have reached surface areas of more than a hundred square kilometres through lake expansion and merging with other lakes (Weller and Derksen, 1979; Grosse et al., 2013). Lake areas can also display seasonal changes due to substantial snow-meltwater influx, full or episodic connectivity to fluvial systems, or seasonal changes in the evaporation-precipitation ratio (Cooley et al., 2019; Mullen et al., 2023).

Many of the regions with a high number of lakes are lowland regions, typically with a ground ice content of 30% or more (Jones et al., 2022). Among others, they include the large arctic river deltas, the Alaskan north slope as well as the coastal lowlands of the Kara, Laptev, and the East Siberian seas (in 't Zandt et al., 2020). Some of these areas are underlain by Yedoma deposits that originated in the late Pleistocene (Strauss et al., 2017). Despite being especially ice-rich, Yedoma deposits also have an especially high organic carbon content ranging from 2% to 5%, which adds to the relevance of thermokarst

lakes for global carbon emissions (Walter Anthony et al., 2018; Freitas et al., 2025). The accelerated permafrost degradation underneath them can make them significant sources of carbon. Largely anaerobic sediment conditions in thermokarst lakes facilitate biodegradation of organic carbon that produces methane, an especially potent greenhouse gas (e.g. Heslop et al., 2020). Consequently, thermokarst lakes often display larger methane emissions in comparison to other types of Arctic lakes or the surrounding land surface (Wik et al., 2016; Walter Anthony et al., 2018).

Thermokarst lakes can alter local and global climate not only through their impact on the carbon cycle. They also influence energy and water fluxes between land surface and atmosphere as well as surface and subsurface hydrology. Waterbodies typically lead to higher evaporation rates, decreased surface roughness and enhanced diurnal and seasonal cycles of heat storage and release by the land. By altering water storage and drainage networks, they can impact the timing of surface runoff into rivers, streams and the ocean. Using the MPI Earth System Model (MPI-ESM), de Vrese et al. (2023) showed that such

changes to energy and water cycling could have effects beyond permafrost regions through biogeophyscial teleconnections. The main biogeophysical effect of a drying Arctic is likely a decrease in summertime cloud cover. This so-called permafrost cloud feedback could amplify global warming (de Vrese et al., 2024).

In turn, local climate influences the behaviour of thermokarst lakes. However, it is not entirely clear how their distributions will develop under global warming. Several studies have tried to obtain past trends of lake areas in regions across the Arctic

using satellite images, with studies observing both negative (Jones et al., 2011; Nitze et al., 2017; Lara and Chipman, 2021) and positive (Nitze et al., 2017; Lara and Chipman, 2021; Veremeeva et al., 2021; Luo et al., 2022; Zhou et al., 2024) trends. Recent satellite data analysis by Webb et al. (2022) and Chen et al. (2023) have found a large-scale drying trend since 2000 across lake-rich regions in the northern permafrost domain. However, it has been questioned that coarse, kilometre-scale remote sensing data, as used by Webb and Liljedahl (2023), is capable of correctly representing regional scale landscape developments (Olthof

et al., 2023; Webb et al., 2023a). According to results from Webb and Liljedahl (2023), decreasing lake area is especially





dominant in discontinuous permafrost zones, while continuous permafrost zones show similar numbers of sites with increasing and sites with decreasing lake area. The study also found no proof of a relationship between precipitation-evaporation balance and lake area trends, thereby supporting the hypothesis that the primary driver of Arctic lake area change is permafrost thaw. Nitze et al. (2020) detected an increase in lake drainage events linked to abundant rain- and snowfall and extremely warm mean

annual air temperatures in Northwestern Alaska, and suggest that this connection could also be explained by permafrost thaw around the lake margins as a result of these weather phenomena.

  Despite the abundance of remote sensing analysis, stand-alone models have been limited to the scale of individual or only a few lakes, even though the region-wide development is of relevance for regional and global climate feedbacks. Ling and Zhan developed a two-dimensional finite element heat transfer model to simulate talik development under thermokarst lakes

(Ling and Zhang, 2003) as well as refreezing of taliks in drained lake basins (Ling and Zhang, 2004). This model, however, treats the lake area as constant. Plug and West's numerical model (Plug and West, 2009) simulates the expansion of a lake in cross-section through thermal processes and mass wasting. It focuses on only one single lake and does not include any site specific processes. Building on this, Kessler et al. (2012) developed a three-dimensional thermokarst lake model also integrating evolution of methane fluxes with growing lake dimensions. Using a three-dimensional Stefan equation, Ohara et al.

(2022) model the thermokarst lake dynamics and talik structure of an individual lake in northern Alaska. Langer et al. (2016) coupled the bulk model FLake (Mironov et al., 2003; Kirillin et al., 2011; Boike et al., 2015), that simulates thermal dynamics in waterbodies, with the permafrost land surface model CryoGrid3 (Westermann et al., 2016), resulting in a one-dimensional land surface model including a simplified representation of lateral waterbody growth, that depends on site specific parameterizations and assumes constant lateral growth. While being useful for individual lake settings, these mentioned models are not designed

to capture the diversity of individual lake behaviour and interactions across a landscape.

  The lack of pan-Arctic models can be ascribed to the fact that thermokarst lakes are results of small-scale processes that depend on meter-scale surface and sub-surface heterogeneities, which cannot be resolved in larger scale models. This scaling gap leads to a poor representation of permafrost-related processes and possible biases in climate models and ESMs. Even in light of current efforts to increase resolutions of ESMs, a deterministic and physics-based representation of thermokarst

lake dynamics on the landscape or pan-Arctic scale remains challenging due to the lack of area-wide high-resolution data on the relevant soil conditions, particularly ground ice distribution. While significant advances in high-resolution remote sensing of the land surface have been made, such as the mapping of ice wedge polygons (Liljedahl et al., 2024), the detection of sub-surface heterogeneities and properties is still difficult.

  To simulate thermokarst lake changes over areas of hundreds of kilometres, van Huissteden et al. (2011) tried to bridge the

scaling gap with a stochastic modelling approach. They assume lake change processes to relate linearly to the deviation from a reference climate, specifically annual precipitation, July temperature and mean annual air temperature. The model is grid-based and initialized with random fractions of ice content for each grid cell following a normal distribution. It therefore requires some prior information on maximum ice content and ground ice distribution in the simulated region in order to accurately represent these conditions. The cells in which thaw is happening are selected randomly, however cells with a large ice content are

preferred. Drainage is only included through lakes expanding onto a prescribed drainage network consisting of rivers, making



the simulation outcome highly depended on these network representations. Other various forms of drainage, such as internal drainage through sub-surface drainage channels, are not included.

In a study of landscape-scale thermokarst lake evolution by Victorov et al. (2019b), lake formation and expansion are also considered to be stochastic. Based on four cases with different assumptions on these processes, theoretical statistical distribu-
tions for lake number and lake areas were derived analytically and empirically verified at 16 sites. For all cases, no drainage is assumed to happen. Under this assumption, the study finds that the number of lakes follows the Poisson distribution on an area that is relatively homogeneous in terms of geomorphological and geocryological landscape conditions. The distribution of lake sizes differs depending on the assumptions. However, the empirical data for the 16 sites shows a lognormal distribution in the majority of the cases, suggesting that "a model based on the proportionality of the growth rate of thermokarst lakes to
the average heat loss density through the side surface", might be most suitable to simulate changes in lake areas.

In another study, Victorov et al. (2019a) considered the development of drained lake basins, often called "khasyreis" in Russian literature, and derived a theoretical statistical distribution of their sizes with a similar analytical approach. Assuming that the distribution of fluvial drainage sources is assumed to correspond to a Poisson distribution and following the same assumptions about lake formation and expansion as in Victorov et al. (2019b), they find that the average radii and diameters
of drained lake basins should follow the Rayleigh distribution, while their number across an area corresponds to a Poisson distribution. In Victorov et al. (2023), they calibrated their model and further modified it using meter-scale satellite imagery for different thermokarst sites in Siberia. Here, they also differentiate between thermokarst lakes that have formed on undisturbed area and secondary lakes that have formed in drained lake basins, and found that the observed size distributions for both mostly fit with theoretical integral-exponential distributions, but have different distribution parameter. They conclude that formation,
expansion and drainage of thermokarst lakes is in dynamic equilibrium. Their approach does not directly consider influences of changing climate variables, but rather focuses inherent lake dynamics under current conditions and drainage due to pre-existing fluvial erosion networks, and is therefore not entirely suitable for projections of thermokarst landscape developments into the future.

In Nitzbon et al. (2020) CryoGrid3 was extended beyond the one-dimensional site level by using laterally coupled tiles.
While this tiling approach allows for a representation of spatial heterogeneity in surface and subsurface conditions, pan-Arctic or landscape scale simulations would still require an accurate initialization of these heterogeneities, including the spatial distribution of ground ice. The one-dimensional and physics-based nature of CryoGrid3 also suggests that a high resolution would be necessary for reliable results with such large scale simulations, making it computationally expensive.

In light of the limitations of previous models for the ESM scale and in part based on the assumptions by Victorov et al.
(2019b), we propose a stochastic modelling approach, where formation, expansion and drainage are considered to be probabilistic across a region and simulated using common stochastic processes. We represent formation and abrupt drainage with two independent Poisson processes that calculate the number of new and disappearing lakes per time step, respectively. For the simulation of variations in individual lake areas, we use Geometric Brownian Motion, which thereby represents lake expansion and gradual drainage. With this method, it is possible to parametrize the stochastic components of lake dynamics and
not only capture possible trends in lake areas, but also their volatility. We conduct three idealized simulations to explore the





dynamic regimes that our model is able to capture, and test our approach by parameterizing with remote sensing data products containing a 20-year timeseries of individual annual lake areas.

## 2 Method

### 2.1 Model Framework

Our model is written in python and has an object-based approach. A diagram of the modelling scheme can be seen in Figure A1 The model represents lakes as circular objects in a 2D plane, thereby only considering a lake's surface area. While thermokarst lakes can have different shapes depending on the region and geomorphology (e.g. Grosse et al., 2013), we regard a circular representation to be an appropriate simplification for a first approach, since they can be expected to radially expand under the assumption of flat terrain with broadly uniform ice content. The 2D approach is based on the simplified assumption that

thermokarst lakes have similar shapes that can be deduced from their surface area, and puts the focus on land surface and atmosphere exchanges while keeping the model simple enough for computational limitations of a potential implementation into ESMs.

Each lake is defined by the coordinates of its centre and it's surface area. The coordinates are randomly chosen from a uniform distribution, meaning that each lake initiation location is randomly distributed in space. The lake's location is relevant

for the model's merging algorithm. At each time step, the model will check for overlapping lakes and merge them by adding the area of the smaller lake to the larger lake and calculating a new centre of mass. The smaller lake is then deleted from the lake inventory. We use this simplistic merging algorithm to keep the model from creating overlapping lake areas and as a first step to account for lake merging without resolving spatial surface and sub-surface heterogeneities.

We use a time step of one year, thereby disregarding seasonal changes in lake area. The lakes' number $n$ and annual surface

areas $a_{i=1,...,n}$ change over time based on the following approaches.

Victorov et al. (2019b) assume lake formation to be probabilistic and to occur independently on disjoint sites. This assumption of independence is in line with the fact that thermokarst happens due to a localized thawing process that depends on soil conditions at that particular point that are not directly influenced by thermokarst at a different point. When additionally assuming that the formation of the initial thermokarst depressions took place during a relatively short time or approximately at the

same time, it can be shown that the number of depressions in an area obeys the Poisson law, meaning that the probability of $k$ thermokarst depressions appearing during one year in $A_f$, which is the area available for formation, is

$$P_f(k, A_f) = \frac{(\lambda_f A_f)^k}{k!} e^{-\lambda_f A_f}, \tag{1}$$

where $\lambda_f$ is the formation rate, which corresponds to the average number of depressions per unit $A_f$ and year. An analysis of observational data from 16 sites showed strong agreement between the distribution of lake numbers and the Poisson distribution

(Victorov et al., 2019b).





When considering a continuous possibility of depression formation over time, instead of only synchronous formation, Victorov et al. (2019b) assume that the formation occurs independently in different time intervals. An argument for the temporal independence of thermokarst events is the random nature of triggers, such as extreme weather or disturbances through wildfire. Following these assumptions, lake formation can be simulated using a Poisson process. Following Pinsky and Karlin (2011), a
Poisson process fulfils the following conditions.

1. For any time point, the process increments are independent random variables.

2. For $s \geq 0$ and time $t > 0$, the random variable $X(s+t) - X(s)$ has the Poisson distribution (eq. 1).

3. The random variable $X(0)$ has the value 0.

We translate this idea into our computational model by implementing an operation that picks the number of new lakes at
each time step from a Poisson distribution using the function *numpy.random.poisson*. The model gets initialized with an array containing a maximum number of possible lakes with an initial area of zero. At each time step, the picked number of new lakes gets activated from the initialization array and each of them get an initial area of 1 $km^2$. From then on, their indices will be tracked in an array for active lakes and they will be subject to annual lake area changes.

Victorov et al.'s (2019b) analytical derivation of lake size distribution is based on the assumption that lake growth happens
independently for individual lakes and that it is proportional to the heat reserves in the lake and inversely proportional to the lateral area of the lake basin. This relationship follows when assuming that the growth rate is proportional to the heat loss through the side surface of the lakes, which is the process that leads to further permafrost degradation in the surrounding area and thereby to lake expansion. From this, they derive the following equation for the increment of the lake radius $\Delta r_j$ for the $j$-th year.

$$\Delta r_j = \epsilon_j r_j \tag{2}$$

Here, $r_j$ is the lake radius at year $j$ and $\epsilon_j$ can be written as

$$\epsilon_j = \frac{\alpha c t°}{2} \epsilon_j^0 \tag{3}$$

with $c$ denoting specific heat, $t°$ the average temperature of water, $\alpha$ the share of the heat amount in the water mass that leaves through the lateral area of the lake basin, and $\epsilon_j^0$ a random variable that takes the impact of episodic factors into account.
These factors can include the thickness of the snow cover, the volume of storm runoff, soil temperatures, precipitation and other variables. Since the values $\epsilon_j$ are independent, the central theorem can be applied, which says that the sum of a large number of independent items is normally distributed. It follows that the growth process of thermokarst radii can be considered as a Markov random process with the following transition function (Victorov et al., 2019b), assuming that the future behaviour only depends on the current state and not on the past (Pinsky and Karlin, 2011; Ibe, 2013):





$$f(v, r, t) = \frac{1}{\sqrt{2\pi}\sigma r \sqrt{t}} e^{-\frac{(\ln \frac{r}{v} - \mu t)^2}{2\sigma^2 t}}. \tag{4}$$

The initial radius of the thermokarst depression is denoted with $v$, while $\mu, \sigma$ are the distribution parameter and $r$ is the depression radius at time $t$. Assuming that the initial size $v$ is one unit radius, it can be derived that the radii have a lognormal distribution at any time $t$ (Victorov et al., 2019b):

$$f_r^0(r, t) = \frac{1}{\sqrt{2\pi}\sigma r \sqrt{t}} e^{-\frac{(\ln r - \mu t)^2}{2\sigma^2 t}}. \tag{5}$$

The equation still holds true when exchanging the lake radius $r$ with the lake's surface area $a$, because these two variables have a quadratic relationship. Agreement of observed lake sizes with this distribution was found in 14 out of the 16 sites that were used for empirical verification (Victorov et al., 2019b).

To model a lognormally distributed variable such as the lake area, the stochastic process of Geometric Brownian Motion (GBM) can be used, which fulfils the condition of a Markov process and is a continuous-time stochastic process, in which the logarithm of the random variable follows Brownian Motion (BM) (Ibe, 2013). BM is a "random walk" process that models random continuous motion (Ibe, 2013). GBM is defined by the stochastic differential equation (SDE)

$$dY(t) = \mu Y(t) dt + \sigma Y(t) dB(t) \tag{6}$$

with $Y(t) = e^{X(t)}$, where $X(t)$ is a BM with drift $\mu$ and can be written as $X(t) = \mu t + \sigma B(t)$ (Ibe, 2013). The parameter $\sigma$ is the volatility and $B(t)$ is a standard BM, which has a mean of zero. The drift parameter $\mu$ alters the process so that there can be an overall trend in the behaviour of the random variable. It can also be called the "deterministic component" of the process, whereas the volatility parameter is the "stochastic component" (Ibe, 2013), as it is a measure of the process' randomness. It can therefore be viewed as a representation of the random variable $\epsilon_j^0$ from eq. 2.

To compute the individual lake area $a_i(t)$ at time $t$, we use the SDE's solution:

$$a_i(t) = a_i(t-1) e^{(\mu - \frac{1}{2}\sigma^2)t + \sigma B(t)}, \tag{7}$$

where $i = 1, ..., n$

We include the BM term $B(t)$ in our model using the *numpy* function *random.normal*, which randomly chooses a value from a normal distribution. In our case, we use time step $dt$ of our simulation as the standard deviation for the normal distribution.

Since GBM can be described as a form of exponential growth with volatility, it is also in line with the mentioned assumptions on lake expansion from Victorov et al. (2019b). For lakes that are assumed to be cylindrical objects with the same shape, these assumptions can be simplified to a rate of change of lake surface area and a coefficient of proportionality that includes a random variable taking into account the impact of episodic factors as in eq. 2.





We treat gradual and abrupt drainage as two separate processes, since we assume them to have different triggers. We include gradual drainage in the GBM by allowing the drift parameter $\mu$ to be negative (eq. 7). This way, lake areas in the model can

decrease from one year to another not only due to volatility but also due to changes in the sign of the drift. This type of drainage is often driven by slower processes such as evaporation, slow creation of sub-surface drainage channels as a result of seepage through the active layer, sedimentation or vegetation growth within the lake. Abrupt drainage, on the other hand, is triggered by more sudden events such as bank erosion or connection with an existing topographic drainage gradient, such as a river or a neighbouring already drained basin. In our case, we define abrupt drainage as a drainage event, during which a lake drains

completely in less than a year. We use another Poisson process to simulate this

$$P_d(k, A_d) = \frac{(\lambda_d A_d)^k}{k!} e^{-\lambda_d A_d}, \tag{8}$$

with $\lambda_d$ being the abrupt drainage rate and $A_d$ the area, by which $\lambda_d$ is scaled. $A_d$ can either consist of only surface water across the region or the sum of water and drained area. With the scaling, we make sure that the drainage rate is only above zero when lakes are present in the landscape, and that the drainage rate increases the more populated with lakes or depressions the

area becomes. Since it is not clear, which of the two options for $A_d$ is more realistic, we implement two different variants that also concern the scaling of $\lambda_f$. When a lake is completely drained, the model saves its index in a respective array for drained lakes and deletes it from the array for active lakes, to which GBM is applied. It also tracks any area that used to be filled with water and classifies it as "drained area". While the majority of this area is likely not susceptible for new thermokarst as long as no new ground ice has accumulated underneath, a new lake could still form on it due to water pooling in the existing depression

and the basin margins might still contain some ground ice that has not been fully depleted, leading to a particularly high chance of renewed lake formation there. Also, an only partially drained lake could still have an increase in surface water due to high inflow. This makes it difficult to decide how to handle the influence of drained area on lake dynamics in the model. While the model always makes it possible for partially drained lakes to grow again, the possibility of new lakes forming on drained area depends on which model variant is used. In Variant 1, both the sum of water area $\overline{A_{water}}$ and the sum of drained area $\overline{A_{drained}}$

across the region are subtracted from the overall area of the simulated region $A$ to obtain $A_f$ and both are added together to obtain $A_d$:

$$A_{f,v1} = A - (\overline{A_{water}} + \overline{A_{drained}})$$
$$A_{d,v1} = (\overline{A_{water}} + \overline{A_{drained}}). \tag{9}$$

In Variant 2, we only consider the $\overline{A_{water}}$, essentially counting $\overline{A_{drained}}$ towards $A_f$ and neglecting it for $A_d$:

$$A_{f,v2} = A - \overline{A_{water}}$$
$$A_{d,v2} = \overline{A_{water}}. \tag{10}$$



The influence of water and drained area is only incorporated implicitly. The areas $A_f$ and $A_d$ influence the number of formation or drainage events across a region, but not the spatial distribution of new lakes. It is therefore possible to see new lakes forming on drained area in both variants. In Variant 1, however, more drained area will lead to a smaller likelihood of lake formation and a higher likelihood of lake drainage. This effect does not exist in Variant 2, because in this model version these likelihoods are only dependent on the water area.

We introduce an area fraction limit $A_{lim}$, which can stop lakes from growing any further. This limit represents the fraction of the study area that can theoretically be populated by lakes. In Variant 1 this includes drained lakes, meaning that the ratio of $A_{d,v1}$ (eq. 9) and the study area needs to reach the limit for lakes to stop growing. In Variant 2, only the water area fraction influences lake dynamics. It is therefore the ratio of $A_{d,v2}$ (eq. 10) to the study area, that needs to reach the limit. Values for the limit should be determined separately for different landscapes in the Arctic, considering the average ground ice content,

topography and other landscape conditions. This task, however, is not trivial. For the purposes of this work, we use a fraction limit of $A_{lim} = 1$ as default, essentially assuming that 100% of any simulated area can become covered with surface water.

Lake dynamics are influenced by environmental and climatic conditions. We therefore assume that our parameters (formation rate $\lambda_f$, abrupt drainage rate $\lambda_d$, drift $\mu$ and volatility $\sigma$) have different values for different geocryological regions/landscapes and that they can change with a climate forcing. The latter dependence can be accounted for by implementing them as a

function of a set of climate variables.

### 2.2   Parameterization Approach

The values for drift $\mu$ and volatility $\sigma$ as well as formation $\lambda_f$ and drainage rate $\lambda_d$ can theoretically be estimated using time series of observed lake areas. For the estimation of $\mu$ and $\sigma$, it is necessary to calculate the logarithmic returns $R_{log}$ for each lake, i.e.

$$R_{log,i}(t) = ln\left(\frac{a_i(t)}{a_i(t-1)}\right). \tag{11}$$

The volatility $\sigma$ can then be estimated by taking the standard deviation across all lakes for each timestep in order to obtain a time series of $\sigma$. For this, we use the function *np.nanstd*, which calculates sigma with equation

$$\sigma(t) = \sqrt{\frac{\sum_{n=1}^{N}|R_{log,n}(t) - \overline{R_{log,n}(t)}|}{N}}, \tag{12}$$

where $N$ denotes the number of values and $\overline{R_{log,n}(t)}$ the mean of $R_{log,n}(t)$. That mean can be used to get an estimate for

$\mu(t)$, after correcting it for the negative effect on returns that is being exerted by $\sigma$:

$$\mu(t) = \overline{R_{log,n}(t)} + \frac{\sigma^2}{2}. \tag{13}$$





A timeseries of estimates for formation rate $\lambda_f$ can be obtained by simply identifying and counting lakes that had a water area of zero until a specific year. For the drainage rate $\lambda_d$ all lake objects can be counted that have a water area of zero from one specific year until the end of the recorded timespan. Because $\lambda_d$ represents the abrupt drainage rate specifically and therefore should not contain lakes that have drained gradually, we only consider lakes that have lost 50% of their maximum water area in one timestep. Depending on which simulation variant we choose, we need to scale formation and drainage rate using either $A_{f,v1}$ and $A_{d,v1}$ or $A_{f,v2}$ and $A_{d,v2}$ according to equations 9 and 10, respectively.

For the implementation of a climate dependence, the correlation between these parameters and climate variables should be investigated and quantified. We assume thawing degree days (TDD) and total annual precipitation $P$ to be the most relevant climate variables. TDD are the sum of the temperatures for every day in a year, where the average daily temperature is above freezing. On these days, it is possible that some of the ground ice starts to melt, thereby initiating ground subsidence and possible lake formation, drainage and lake expansion. High precipitation can also lead to lake formation, lake expansion, or drainage by facilitating permafrost degradation and water pooling. Furthermore it leads to an increased water input into both existing lakes as well as the drainage channel. In order to reduce the stochastic effects, we smooth the timeseries of formation and drainage rate by taking the 3-year running mean of these rates and comparing them to the 3-year running mean of the climate variables.

## 2.3 Simulations

### 2.3.1 Idealized

Our model is able to capture three general dynamic regimes: complete drainage of the landscape (A), oscillating behaviour of water and drained area (B), and stabilization of area fractions (C). To demonstrate these regimes, we perform three exemplary 1000-year simulations with idealized parameterizations, which can be found in Table 1. These parameter values are hypothetical and remain constant, meaning that they are independent of a climate-forcing. They are chosen to create qualitative and demonstrative examples of the regimes within the simulation period of 1000 years with a timestep of one year, and should therefore not be interpreted as realistic representations of observed lake dynamic speed. Lake expansion, for example, can generally be assumed to be slower by about an order of magnitude. Most observed mean expansion rates lie between 0.1 and 2 m/a with some single lakes showing higher rates of up to 6.01 m/a (Jones et al., 2011). Since the parameters are hypothetical, the simulations could also be interpreted as simulations over a 10 ka year time span with a timestep of 10 years. In that case, the parameter values in Table 1 would have to be considered to be per decade and not per year. All three of the simulations start with a landscape without lakes or freshly drained lake basins. The simulation area is 40x40 km. This size is representative of a grid cell size for the Arctic in the Max Planck Institute's land surface model JSBACH/ICON-Land (Jungclaus et al., 2022) with R2B6 resolution.



**Table 1.** Hypothetical parameter values per year, fraction limit and model variant used for the three different idealized simulations.

| Regime / Parameter | $\lambda_{f,v1/v2}$ $[m^{-2}]$ | $\lambda_{d,v1/v2}$ $[m^{-2}]$ | $\mu$ | $\sigma$ | $A_{lim}$ | variant |
|---|---|---|---|---|---|---|
| A. Complete Drainage | 1e-9 | 1e-10 | 0.3 | 0.02 | 1 | 1 |
| B. Oscillation | 1e-9 | 1e-10 | 0.3 | 0.02 | 1 | 2 |
| C. Stabilization | 1e-8 | 1e-12 | 0.03 | 0.01 | 0.5 | 1 |

### 2.3.2 Observation-based

We also test our approach by applying the parameterization method to a dataset of lake areas (Nitze and Nicholson, 2025) derived from time-series of the JRC Global Surface Water Dataset (Pekel et al., 2016), which was itself derived from the multi-temporal orthorectified Landsat 5, 7, and 8 archive data. Pekel et al. (2016) classified the pixels in these satellite images as open water, land, and non-valid using expert systems, visual analytics, and evidential reasoning. As part of a larger effort to create a pan-Arctic lake change dataset, we calculated lake footprints based on the methodology by Nitze et al. (2017, 2018) for the Arctic part of UTM Zone 54 N and an extended time series from 2000 to 2021 (Nitze and Nicholson, 2025). That is, for each lake larger than 1 ha, lake polygon footprints were obtained as shown in Fig. 1. Then the annual surface water areas within each lake polygon were extracted from the JRC Global Surface Water Dataset for the period 1984-2021 using Google Earth Engine, employing the geemap (Wu, 2020) and eemont (Montero, 2021) Python packages. Within each polygon, the fraction of permanent water area, land area, no-data area, and seasonal water area for each year were extracted. However, lake area values before 2000 were not reliably available due to missing or sparse Landsat observations.

As an additional aid, we used a dataset by Chen et al. (2023), who identified lake drainage events across the Arctic using an object-based analysis assisted by the JRC surface water data product as well as another surface water product published by the Global Land Analysis and Discovery (GLAD) team (Pickens et al., 2020). Both products are derived from the archive of 30m-resolution Landsat imagery. For their analysis, Pickens et al. (2020) defined drainage events as any time a lake lost more than 50% of its area and was initially larger than 1 ha. They identified the main year of drainage for each event. Due to the discontinuous nature of the Landsat data, clear identification of lake drainage could only be done for the years 2000-2020, which is the period that we focus on.

Our study area (Fig. 1) encompasses about 48623 km$^2$ and lies in the coastal region of the Yana-Indigirka Lowland in Siberia. We chose this test region for its high density of thermokarst lakes and basins. It spans the northern part of UTM Zone 54 N between 138°E and 144°E, and 70°N and 73°N. According to the ecoregion classification by Olson et al. (2001), the study area can be classified as Northeast Siberian coastal tundra. In the landform categorization of the circum-Arctic Permafrost Region Pond and Lake database (PeRL) (Muster et al., 2017) the area is denoted as alluvial-limnetic with continuous permafrost and about 50 % ground ice content. It is underlain by Yedoma deposits and contains 22939 lakes that were identified in the lake area dataset. We also tested our approach for a random 40x40 km cell within the region (Fig. 1), representing a grid cell size for the Arctic in JSBACH / ICON-Land with R2B6 resolution in line with the idealized simulations in Sect. 2.3.1 and 3.1. This



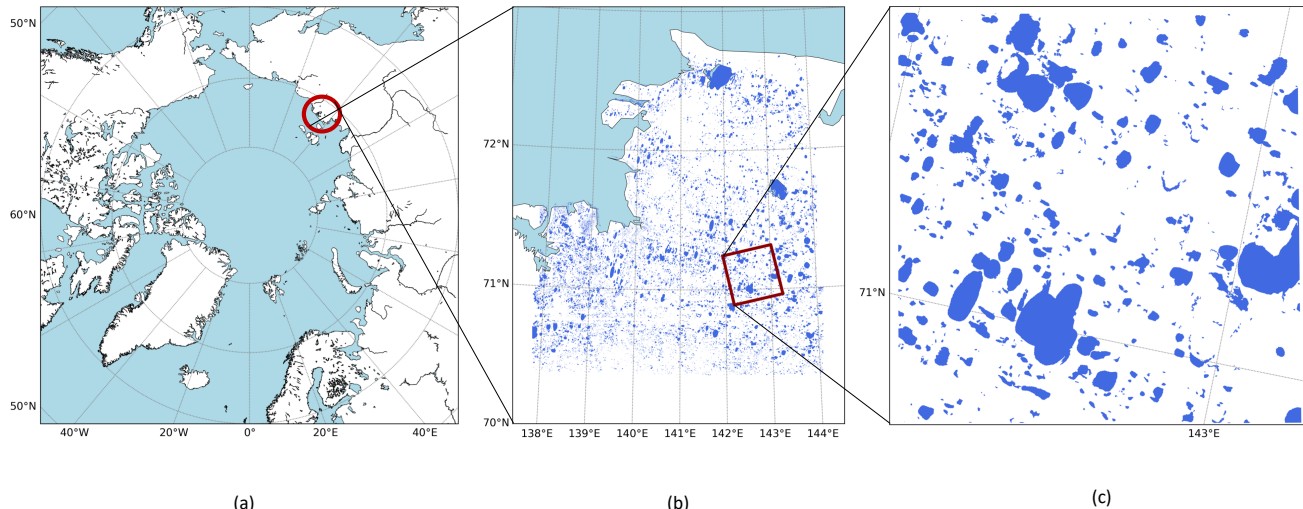

**Figure 1.** Study region with location of example lake change dataset. An Arctic stereographic projection map shows the location of the study region in Eastern Siberia at the coast between Laptev and East Siberian Sea (a). The study region and calculated lake footprint dataset encompass the northern part of UTM Zone 54. The lakes from the dataset are shown as blue polygons (b). We focus on a random 40x40 km cell within that region (c).

cell contains 522 identified waterbodies. Before applying the parameterization method to the lake area data, we exclude all
360 values for polygons and years where any fraction of the area was classified as no-data.

After obtaining parameter estimates with the method described in Sect. 2.2, we investigated whether there was a quantifiable relationship between the stochastic parameters and climate variables thawing degree days (TDD) and total annual precipitation $P$ (Fig. 5a) between 2000 and 2019. We obtained the climate data from reanalysis data from the Global Soil Wetness Project Phase 3 (GSWP3) (Lange et al., 2021), and calculated TDD from the daily temperatures. We then tried to find correlations
between the parameters and the climate variables by calculating the Pearson and Spearman correlation coefficients between the two timeseries. For these calculations, we used the *corr* method from the python package *pandas*. As it is not clear how much the system lags in response to the forcing, we checked the correlation between parameter estimates and climate variables in the same year as well as between parameter estimates and climate variables in the previous year.

Instead of deriving a climate-dependent function for each parameter, it is also possible to calculate the parameter as constants
by taking the mean of the calculated timeseries, thereby assuming them to be stable over that time period. For the calculation of $\mu$ and $\sigma$, we first created the rolling means for 3-year windows to get more robust results. The constant rates $\lambda_f$ and $\lambda_d$ are calculated using the counting method described in Sect. 2.2 and dividing the sum of counted lake objects by the years in the timeseries. As an example, we tested this approach in the 40x40 km cell, chosen for its representativeness of a typical land surface model resolution and its relative computational inexpensiveness.





# 3 Results

## 3.1 Idealized Simulations

In the following, we discuss the idealized simulations as described in Sect. 2.3.1 and show timeseries of simulated water and drained area as well as lake number. For each regime, we also include three examples of spatial plots that show the active and drained lakes in the system from a bird's eye view at different times of the simulation. These spatial plots should be seen as visual aids to understand the lake distributions, rather than realistic representations of the simulated landscapes, since lake surfaces are simplified as circles in our model and the boundaries of the available landscape only affect area fraction implicitly. This is why lakes can be seen growing beyond the area's boundaries, even though overall water or disturbed area fraction is limited to a value of 1 or lower in all three simulations.

### Regime A: Complete Drainage

For a complete drainage regime, we use Variant 1. As described in Sect. 2.1, this variant's lake formation will become less likely the more the landscape is filled with water and drained area. At the same time drainage probability will increase. As a consequence, the system will likely reach a point where every lake has drained and no new lakes can form. In Fig. 2a an initial increase in water area fraction in the study region and number of lakes can be seen, as first lakes form in the previously lake-free system. In year 100 of the simulation, a lake-filled landscape has established and first lakes have started to drain. A spatial representation of this can be seen in Fig. 2b. Newly formed lakes have been randomly distributed in the simulation area of 40x40 km. Since $\mu$ is positive, lakes have also expanded, causing them to reach different sizes and occasionally merge. With more larger lakes, abrupt drainage is more likely to result in stepwise decreases of water area fraction, as can be seen in Fig. 2a between year 240 and 241. In year 240 the system is dominated by one large lake (Fig. 2c), which completely drains in year 241. Lakes continue to drain until the water area fraction has reached zero. Since lake formation probability is scaled by $A_{f,v1}$ according to eq. 8 and 9, almost no new lakes can form after the drained area has reached a certain threshold. This leads to a relatively stable system, in which almost all area is covered by drained basins (Fig. 2a & 2d). While there are no known regions that are completely covered by drained lake basins, but do not have any extant lakes, there are areas with an especially high fraction of drained area, namely the Yamal Peninsula in Siberia (von Baeckmann et al., 2024) with 78% and the Yukon–Kuskokwim Delta (Jones et al., 2022). It can not be ruled out that the remaining lakes in these regions will also drain in the future, without new lakes forming.

### Regime B: Oscillation

In order to achieve a regime, in which water and drained area oscillate with an inverse correlation, it is necessary for new lakes to be able to form on already drained area. We therefore use Variant 2 for this regime. Within the first 268 years/iterations of the simulation, the system behaves similarly to regime A, while first lakes form and expand. Eventually, some become affected by abrupt drainage (Fig. 3a). The resulting landscape in year 100 (Fig. 3b) resembles the landscape at the same time of regime A in



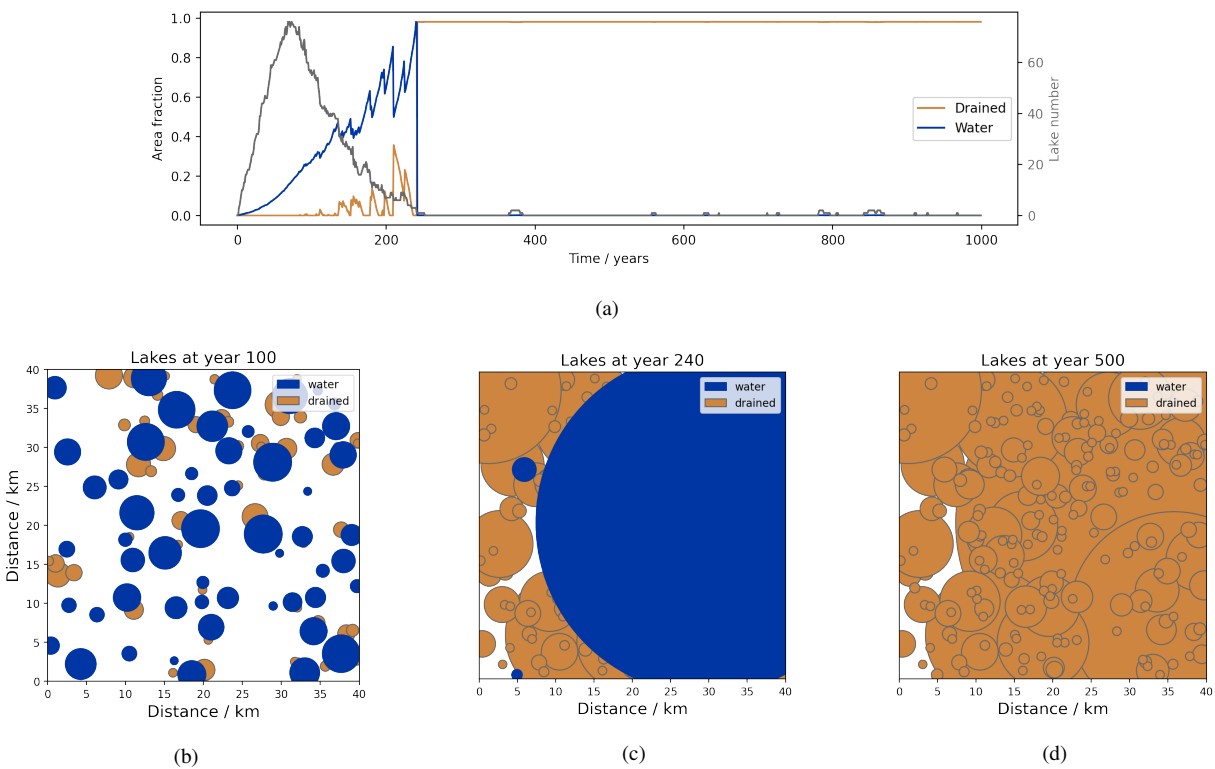

**Figure 2.** Model output for an idealized simulation for a complete drainage regime ("Regime A") using the parameter values from Table 1: Timeseries of water and drained area fraction as well as lake number (a) and spatial representation of lake distribution and drained area extent for simulation year 100 (b), year 240 (c) and year 900 (d). Water area is displayed in blue and drained area in light brown.

Fig. 2b. A similar complete drainage regime as described for regime A happens between year 266 and 267. In this simulation, however, new lakes eventually appear, since Variant 2 scales the formation probability with $A_{f,v2}$ (eq. 8 & 9), which only contains the water area fraction. This means, that lake formation probability will rise when lakes drain and water area fraction decreases, regardless of large parts of the area being covered by drained basins. This behaviour repeats itself four times within the simulation period, as can be seen in Fig. 3a. Figure 3c represents a snapshot of the system at year 500, where lake expansion and merging have lead to one big lake, which is close to draining and leading to the second of the abrupt system changes due to a stepwise decrease of water area fraction to almost zero. In year 900 (Fig. 3d), a landscape with several newer differently sized lakes that resemble the lake distribution from year 100 is visible (Fig. 3b). However, the lakes in year 900 are already the fourth generation of lakes. This type of landscape, where active lakes have appeared on drained lake basins, has been found in several regions across the Arctic (Bockheim et al., 2004; Jones et al., 2012; Roy-Léveillée and Burn, 2017; Fuchs et al., 2019; Bergstedt et al., 2021). Interpreting the simulations as spanning 10 ka years and having a time step of 10 years, would mean that there are four simulated lake generations over the course of 10 ka years, which is the same order of magnitude that has



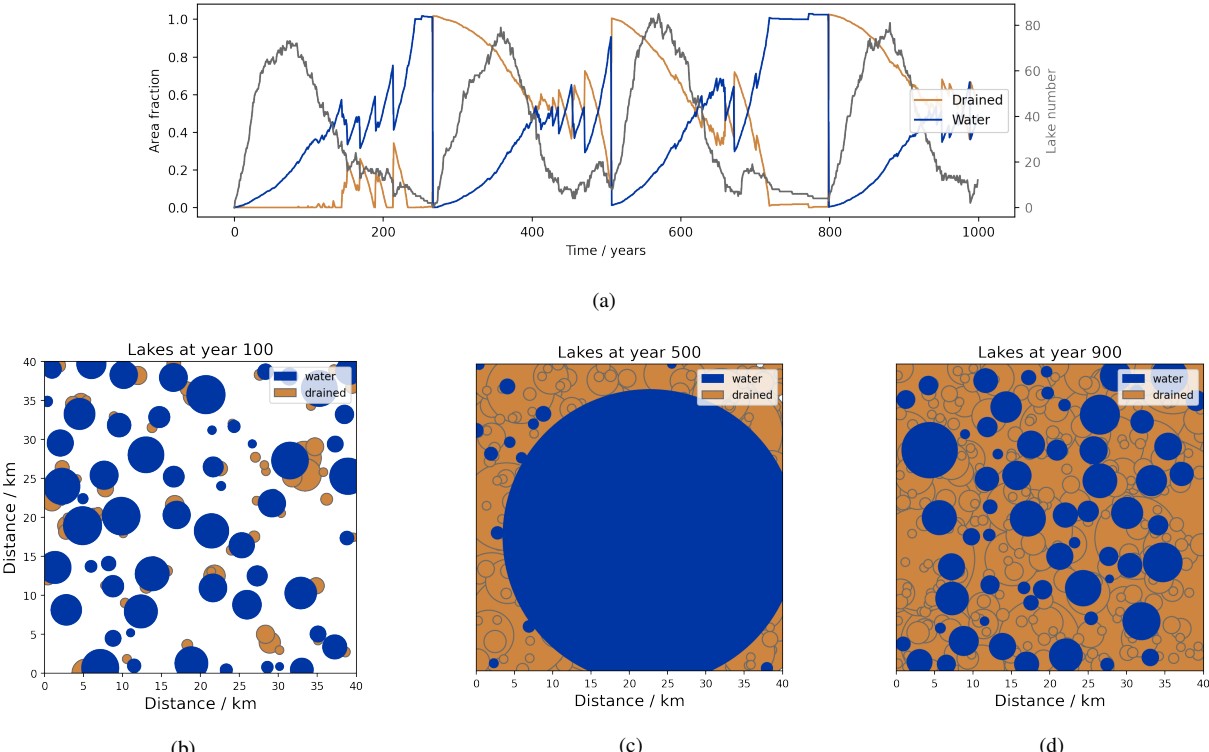

(a)

(b)                              (c)                              (d)

**Figure 3.** Model output for an idealized simulation for an oscillating regime ("Regime B") using the parameter values from Table 1: Time-series of water and drained area fraction as well as lake number (a) and spatial representation of lake distribution and drained area extent for simulation year 100 (b), year 500 (c) and year 900 (d). Water area is displayed in blue and drained area in light brown.

been observed, for example on the Seward Peninsula, Alaska, were Jones et al. (2012) found six generations of overlapping lakes that have formed over the course of the Holocene with the oldest being roughly 9 ka years old.

### Regime C: Stabilization

Quasi-stabilization of the area fractions can occur when they have reached the pre-determined fraction limit, which keeps lakes from expanding further. For stabilization, drainage and formation rate also need to be chosen in a way that both processes are in balance at the fraction limit or their probabilities are close to zero. In the former case, lakes should not become too big, as abrupt drainage of a large lake would have a higher impact on the water area fraction than drainage of a smaller lake and would lead to more pronounced dips in water area fraction. To achieve a quasi-stabilization, we therefore also decrease $\mu$ compared to the other regimes. The high formation rate, still leads to a relatively steep increase in water area fraction in the beginning of the simulation (Fig. 4a). When the sum of water and drained area fraction reaches the limit 0.5, lakes stop expanding. While lakes continue to form, they are also subject to abrupt drainage. This process happens often enough to counterbalance the water area increase through formation, so that the area fractions oscillate around the fraction limit for the rest of the simulation. Figure 4b





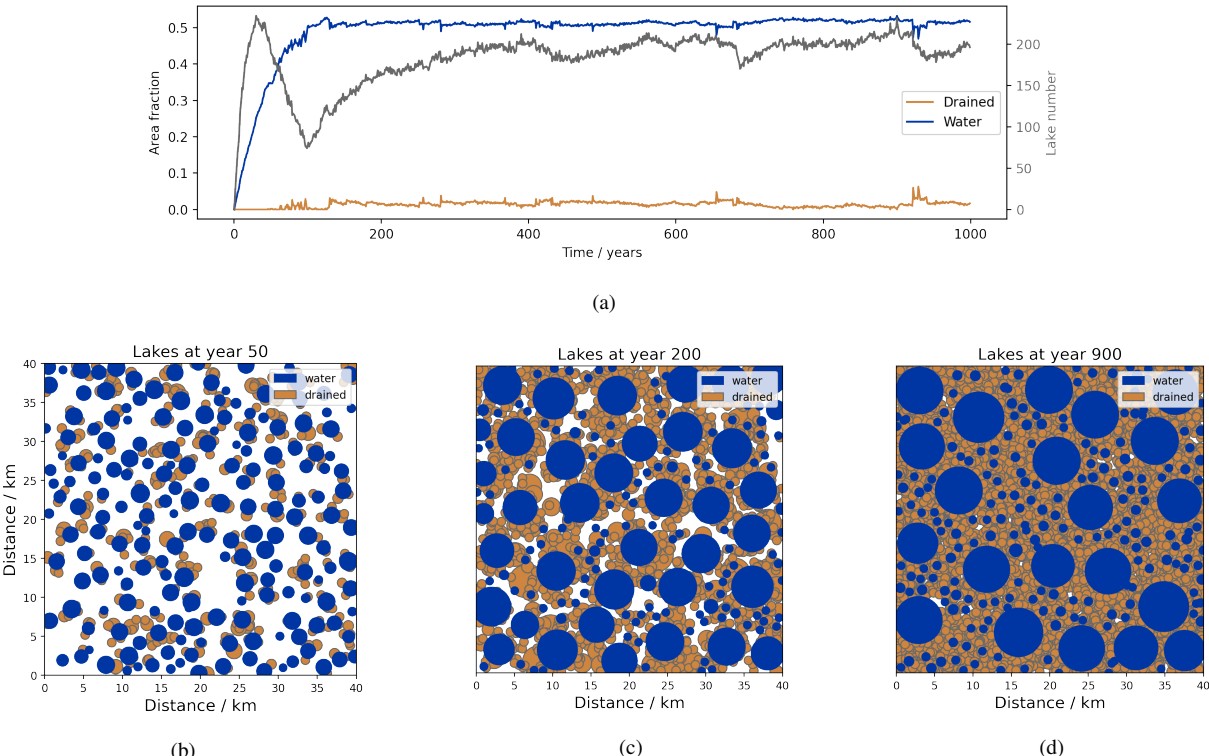

**Figure 4.** Model output for an idealized simulation for a stabilizing regime ("Regime C") using the parameter values from Table 1: Timeseries of water and drained area fraction as well as lake number (a) and spatial representation of lake distribution and drained area extent for simulation year 100 (b), year 500 (c) and year 900 (d). Water area is displayed in blue and drained area in light brown.

shows the landscape at year 50, with many relatively small lakes (Fig. 4c). By year 200 these have expanded and merged into bigger lakes. Additionally, some smaller lakes have formed. The larger size of the older lakes in year 900 (Fig. 4d) can mostly be explained with the merging with newly formed lakes. In this regime, some individual lakes survive the simulation period while only slightly changing size. Thermokarst-affected regions with relatively stable lake area fractions and no large-scale trends over several decades have been identified using remote sensing data (e.g. Jones et al. 2009).

**3.2    Observation-based Simulation**

For the whole study region, we found that 30% of the data points were not useable due to missing pixels in the satellite data, with each year being differently affected and some years having as little as 14% of useable data. Still, we were able to obtain a timeseries of $\mu$ and $\sigma$ with the remaining data points via the method described in Sect. 2.2. We could not confidently identify a lake that has formed within the time period and therefore assume formation rates of zero for each year. Chen et al. (2023)

found 36 drainage events in within the cell between 2000 and 2020. However, some of these events did not lead to a complete drainage of the lake, thereby not completely fitting into our definition of "abrupt drainage". We therefore only considered those



drainage events, for which the authors detected a drainage percentage of 0.9 %, which left 10 events. The area had an average increase of 7.26 TDD over the 20-year period and an increase in $P$ of 0.448 mm (Fig. 5a).

No significant correlation was found between the climate variables and our parameter. The resulting correlation coefficients
for the comparison with the corresponding year's climate are shown in Figure B1a & B1b, and for the previous year's climate in Figure 5d % 5c. The strongest correlation was found between volatility $\sigma$ and previous year's thawing degree days, with a Spearman correlation coefficient of -0.67 (Fig. 5c) and Pearson correlation coefficient of -0.56 (Fig. 5d). This translates into an exponential regression with $R^2$ of 0.45 (Fig. 5b), which we considered to be too weak in order to confidently use this regression function for the volatility calculation in the model. The other correlation coefficients for this analysis were estimated
to be between 0.12 and -0.19 (Fig. 5d & 5c), indicating weak or no correlation. This suggests that either the system has no immediate response to changes in thawing degree days or total annual precipitation during the time from 2000 and 2020, or that the stochastic component is too high to detect it. It is also important to note, that the large data gaps likely influence the robustness of derived parameter estimates and that the drainage rate was calculated based on relatively few drainage events within a 20-year period.

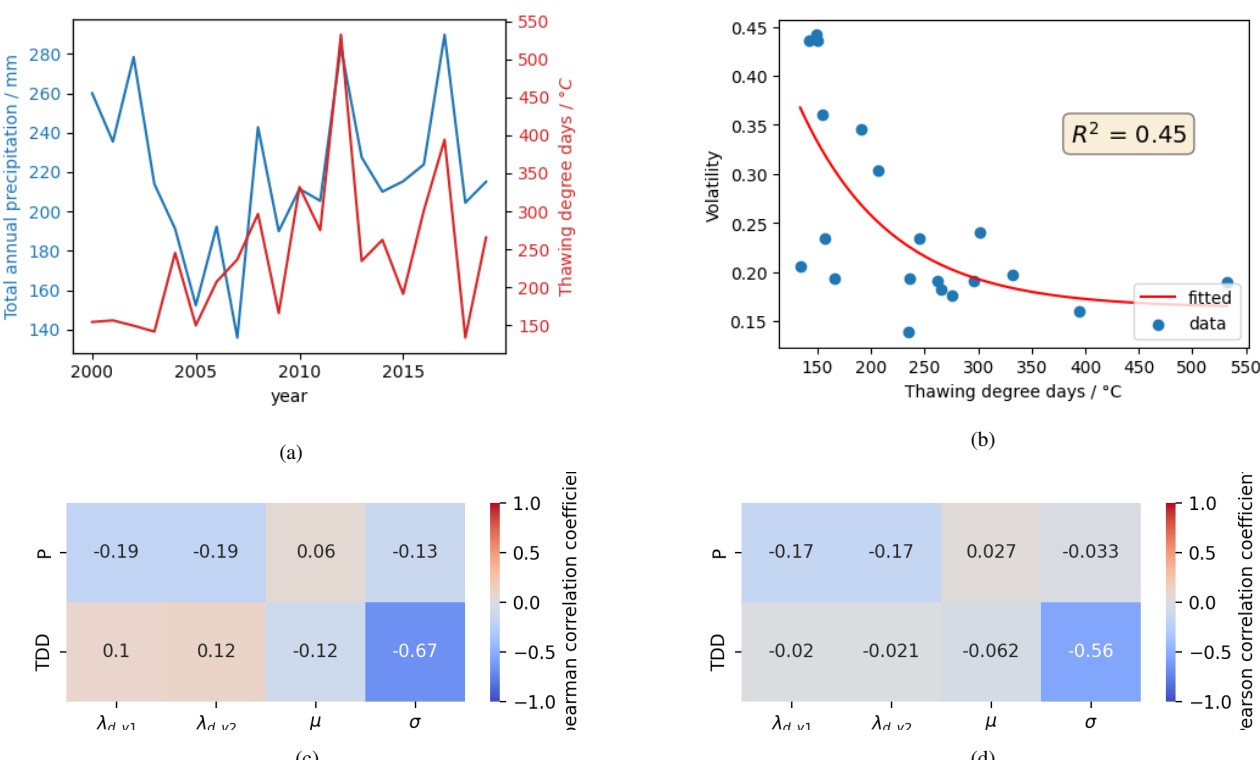

**Figure 5.** Thawing degree days (TDD) and annual total precipitation between 2000 and 2020 in the study region. The data was derived from daily GSWP3 data (Lange et al., 2021). (a), exponential fit between calculated volatility $\sigma$ and TDD of the previous year (b), Spearman (c) and Pearson (d) correlation coefficient between parameters and the previous year's TDD and annual precipitation in the study region between 2000 and 2020.




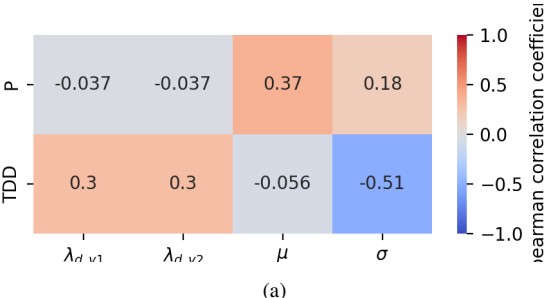
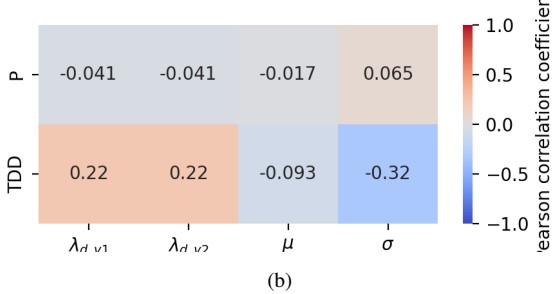

**Figure 6.** Spearman (a) and Pearson (b) correlation coefficient between parameters and previous year's thawing degree days (TDD) and annual precipitation (P) in the 40x40 km cell within the study region between 2000 and 2020.

**Table 2.** Parameter values obtained from remote sensing data products with parameterization method as described in section 2.2.

| Parameter | $\lambda_{f,v1}\ [m^{-2}]$ | $\lambda_{f,v2}\ [m^{-2}]$ | $\lambda_{d,v1}\ [m^{-2}]$ | $\lambda_{d,v2}\ [m^{-2}]$ | $\mu$ | $\sigma$ |
|---|---|---|---|---|---|---|
| Estimated value | 0 | 0 | 1.6670997e-10 | 1.6742647e-10 | 0.0132054 | 0.0787707 |

In the 40x40 km cell, about 33% of data points contain an area fraction that is classified as no-data. Chen et al. (2023) found 1 drainage event in year 2013 that fits into our definition of "abrupt drainage". Similarly to the whole study area, a significant correlation for this subset was found neither for a comparison with the previous year's climate variables (Fig. 6a & 6b) nor the corresponding year's climate variables (Fig. B1c & B1d).

The calculated constant parameter values for the 40x40 km cell are shown in Table 2. It cannot be confidently determined, however, whether these values describe inherent and climate-independent lake dynamics or whether they are specific to the particular climate change scenario. The volatility estimate $\sigma$ was about 5.65 times higher than the drift $\mu$. This suggest that environmental factors that influence individual lake behaviour were the dominant component that drove regional-scale lake dynamics in this area between 2000 and 2020. This is also in line with the low signal-to-noise ratio of 0.37, which we calculated for the mean logarithmic returns (11) across all lakes. When the estimates for $\mu$ and $\sigma$ were put into eq. 7, the term $(\mu - \frac{1}{2}\sigma^2)$ took on the value 0.0101030, which was smaller than the average value of the term $B(t)$. This means that the stochastic term $\sigma B(t)$ would dominate simulated variations in lake area.

We tested the parameterization with ten 100-year long ensemble simulations for Variant 1 and Variant 2, respectively. In these simulations, we essentially assumed that the parameter values do not change during that time period, because either the landscape characteristics and inherent dynamics will continue to dominate dynamics or the trends in climate variables will stay the same as during the parameterization period from 2000 to 2020. We used the lake size distribution from the year 2000 for initialization and distributed the differently sized lakes randomly in the spatial domain.

Figures 7a and 8a show the simulated water and drained area fraction for each member in the ensemble for the two variants, as well as the ensemble mean and the standard deviation. On this timescale, distinct differences between the two variants can





not be determined. The ensemble means of both variants showed an increase of water area fraction. In Variant 1, the fraction

of water area in the study region increased by 228 % from 0.16 to 0.52 during the 100 years of simulation. In Variant 2, water area fraction increased similarly by 227 %, resulting in a water area fraction of 0.53 by year 2100. The yearly growth rate of lake radii averaged 0.45 m/a across all ensemble members for Variant 1 and 0.41 m/a for Variant 2. The standard deviation of water and drained area fraction in Variant 1 temporarily reached values of 0.14 and 0.07, respectively. In Variant 2, they reached similar values of up to 0.15 and 0.05. Ensemble runs with Variant 1 exhibited between 6 and 12 abrupt drainage events

over the simulation period, while Variant 2 experienced 5 - 11. With Variant 1, between 284 and 429 merging events occurred, while Variant 2 led to 268 - 402 merging events, meaning that the lake number decreased accordingly in the respective ensemble simulations. Figure 7b and 8b show the lake size distribution across all ensemble members with mean M and standard deviation SD at the start year 2000, year 2020 and year 2099. With both variants, the distribution flattened slightly and remained skewed to the right, while mean and standard deviation increased. A spatial representation from an example ensemble member for

these three time points can be seen in Fig. 7c to 7e and Figures 8c to 7e. As explained in Sect. 3.1 such representations should be viewed as visual aids only. By year 2020 (Fig. 7d, 8d) some lakes had slightly decreased in size as a consequence of the relatively high volatility parameter $\sigma$. Most, however, had grown or merged. By year 2099 (Fig. 7e, 8e) in the two example simulations one or more especially large lakes had formed that dominate the landscape. The mean drained area fraction in Variant 1 had an average value of 0.05 between years 2086 and 2100 (Fig. 7a), whereas Variant 2 had a drained area fraction

that stayed below 0.03 throughout the simulation (Fig. 8a). This higher value in Variant 1 was mostly due to two ensemble members that showed a sudden increase of drainage area fractions as a result of an abrupt drainage event. This emphasizes the fact that Variant 1 will typically have a higher increase of abrupt drainage probability as the landscape becomes more populated with drained lake basins, because it considers drained area for the scaling of both formation and abrupt drainage rates.

## 4 Discussion

While the works by Victorov et al. (Victorov et al., 2019b, a, 2023) use analytical approaches to derive equilibrium distributions of lake and drained lake basin areas and numbers, our model relies on numerical methods and can represent three different dynamic regimes, depending on parameterization and model variant. Furthermore, it allows for the inclusion of a climate forcing by implementing the model parameters as functions of climate variables besides also capturing internal lake dynamics that result from accelerated permafrost degradation due to the accumulation of water, rather than climate changes.

The model could provide ESMs with information on water and drained area fractions in thermokarst-affected grid cells over time and in response to a climate forcing. With this information, simulated land-atmosphere fluxes of carbon, energy and water could potentially become more accurate. JSBACH/ICON-Land (Reick et al., 2021), for instance, which is the land component of the Max Planck Institute's Earth System Model ICON, divides their grid cells into subgrid-scale tiles that are only defined by their fraction within the grid box. It could use our model output to set the size of lake water tiles. Deriving a simple function

of water area fraction in relation to changing climate variables that emulates our model responses, could provide a way to incorporate thermokarst lake dynamics directly into ESMs without significantly increasing computational costs.





**Figure 7.** Model output for an ensemble simulation with Variant 1 and using the parameter values from Table 2: Time series of water (top) and drained area fraction (bottom) of all ensemble members as well as ensemble means and standard deviation (a). Histograms for size distribution across all ensemble members (b) at start of simulation (left), year 20 (middle) and year 2099 (right) with mean M and standard deviation SD. Spatial representation of lake distribution and drained area extent of an example ensemble simulation for start of simulation (c), year 500 (d) and year 2099 (e) with water area displayed in blue and drained area in light brown.





(a)

(b)

(c)                         (d)                         (e)

**Figure 8.** Model output for an ensemble simulation with Variant 2 and using the parameter values from Table 2: Time series of water (top) and drained area fraction (bottom) of all ensemble members as well as ensemble means and standard deviation (a). Histograms for size distribution across all ensemble members (b) at start of simulation (left), year 20 (middle) and year 2099 (right) with mean M and standard deviation SD. Spatial representation of lake distribution and drained area extent of an example ensemble simulation (c) for start of simulation (left), year 500 (middle) and year 2099 (right) with water area displayed in blue and drained area in light brown.



The three regimes that our model can capture can be expected in thermokarst lake landscapes according to observational studies (Bockheim et al., 2004; Jones et al., 2011, 2009; Roy-Léveillée and Burn, 2017; Bouchard et al., 2017; Fuchs et al., 2019; Bergstedt et al., 2021; Anderson et al., 2019; Jones et al., 2022; von Baeckmann et al., 2024), and result from interactions and feedbacks between lake formation, expansion and drainage and the landscape changes induced by these processes. However, both model variants constitute simplified and incomplete representations of these feedbacks. The development of a more varied topography as lakes form, expand and drain, and its effect on formation and drainage potential is only represented implicitly through the scaling of abrupt drainage and formation probability using water and drained area fractions in the system. It is important to note that this scaling is incorporated linearly, meaning that abrupt drainage probability will increase linearly with more disturbed area in the system as steeper topographic gradients develop, while formation probability will decrease linearly as less area becomes available for formation. These feedbacks could be more complex in reality, however. Furthermore, lake expansion and gradual drainage as well as the fraction limit for disturbed area remain unchanged with landscape changes in the current model version, but these aspects should be taken into account for a more comprehensive feedback representation.

Model Variant 1 represents the fact that there will be no ground ice under recently drained lakes and therefore no new thermokarst, but it does not capture the possible and often observed refilling of drained lake basins with rain or meltwater (e.g. Hinkel et al., 2005), or the presence of remnant ground ice not fully depleted by an earlier lake generation, because new thermokarst lakes are implicitly not allowed to form on drained areas. Once the area of a lake has become zero, the model does not allow it to increase again. This variant therefore does not account for lakes that form due to other processes besides thermokarst. In Variant 2, lake expansion and formation are only limited by the water area in the system. This is equivalent to the assumption that lakes can expand onto drained area and also form on it. This variant thereby could represent lake formation due to re-wetting and new ponding in drained lake basins, which could be caused by water pooling in existing depressions. However, for these type of lakes, the process for lake expansion would not be the accelerated permafrost degradation that is the reason for expansion of most thermokarst lakes. The implementation of lake expansion in the model might therefore not be suitable in these cases. A third option to treat the influence of drained area would be to implement the refreezing of drained lake basins and renewed ground ice formation, which has been observed in many Arctic regions (Jorgenson and Shur, 2007; Lindgren et al., 2012). In this implementation, the drained area would first be considered to be unavailable for lake formation, but become available again after some time. This would essentially represent the processes of refreezing of the taliks in the drained lake basins, the accumulation of new ground ice, and the associated upheaval of the land surface. These processes become especially important when simulating longer timeseries of several hundred years. Early-stage ice wedge polygons have been observed in basins that drained a few 100 years ago (Lindgren et al., 2012). In satellite images, remnant lakes can be found at different locations within a drained lake basin, including the margins, as parts have been lifted up by ground ice accumulation (Jorgenson and Shur, 2007; Jones et al., 2012; Lindgren et al., 2012; Jones et al., 2022). The inclusion of a refreezing algorithm and its parameterization would be challenging, though, since the involved processes are complex and depend on small-scale subsurface conditions. Including the refreezing stochastically could be a way forward.

Our merging algorithm leads to strong decreases in lake numbers, which are likely not realistic. This suggests that the merging implementation is not an accurate representation of real merging. The way that lakes merge in reality is highly case-




dependent and influenced by the shape of the lakes as well as the surrounding landscape and subsurface (Jones et al., 2011).
Actual merging often results in one lake draining into the other without increasing its surface area proportionally. It can also
happen that merged lakes can split again due to partial drainage, which is not possible in the model. The implemented algorithm
is not able to capture this variety of merging dynamics and likely leads to dramatic overestimation of merging events. It can
only represent irreversible and simplified merging, after which the resulting lake has a circular shape and a surface area equal
to the sum of the surface areas of the two previously separate lakes. As a result, some of our simulations yield landscapes with
single lakes of up to 1566 $km^2$, which is unrealistic. Furthermore, the merging algorithm is computationally expensive and
requires lakes to be distributed in space. Altering the model to include a stochastic implementation of lake merging should be
a first step when creating a new model version. Continuum percolation theory includes some approaches, such as the Boolean
model (e.g. Chiu et al., 2013), that could be a basis for such an implementation, but are not directly suitable for our case of
lakes with a varying size distribution. The lake area dataset described in Sect. 3.2 could be an aid in parameterizing a merging
algorithm. Currently this data does not capture lake merging, because lake objects are identified using the largest extent of
connected pixels within a timeseries of raster data. It would be possible to get an estimate for the frequency of merging events
with further processing, however.

The representations of lake expansion and gradual drainage in the model is based on idealized assumptions on thermokarst
feedbacks, which limits the applicability to different kinds of lakes. It might therefore not be suitable for more complex lake
behaviour and lake area changes that are driven by non-thermokarst processes. With an accurate calibration, the stochastic
nature of the model and the inclusion of the volatility term could be able to capture some of this variety in lake dynamics,
however. This is also why the model is not limited by the lack of small-scale subsurface data. The somewhat simple struc-
ture of the model makes it fairly flexible and allows for relatively easy adjustment of the lake process equations as well as
expansion and inclusion of previously neglected aspects, such as ground ice accumulation in drained lake basins or different
merging dynamics. With efforts to increase data availability in permafrost regions, this could allow the model to become more
comprehensive in the future.

The numbers of abrupt drainage events that we simulated using our model with observation-based parameterization are
roughly in line with observed drainage events in Jones et al. (2020), where 98 drained lakes within a 30,000 km$^2$ area between
1955 and 2017 were found, which corresponds to an average drainage rate of 1.58 lakes per year. In our model ensemble simu-
lations we found between 5 and 12 drainage events for a 40x40 km cell over a period of 100 years. Extrapolating these numbers
to a 30,000 km$^2$ this would be an abrupt drainage rate between 0.94 and 2.25 lakes per year. However, the definition of drainage
events in Jones et al. (2020) includes lakes that lost more than 25% of their surface area, whereas we only consider lakes to
be subject to abrupt drainage when they drain completely. While the data from Jones et al. (2020) and our parameterization
data (Nitze and Nicholson, 2025) both cover a coastal plain, the drainage rates could still be specific to the different respective
study regions in Alaska near Teshekpuk Lake and the Yana-Indigirka Lowlands. In Jones et al. (2011), the same definition of
a more than 25% reduction in surface area was used to detect drainage events from high-resolution remotely sensed imagery
from 1950/51, 1978 and 2006/07 for a 700 km$^2$ region on the Seward Peninsula, Alaska. An average drainage rate of 2.3 lakes
per year was found, which is between 50 and 100 times as high as the simulated abrupt drainage rates for our study region and



abrupt drainage definition. In the same study, a mean lake expansion rate between 0.34 and 0.39 m/a were observed, which is only slightly smaller than our simulated mean rates of 0.45 and 0.41 m/a. Observed rates for different regions in the Tibetan Plateau between 1969 and 2010 were even smaller with an average of 0.13 m/a (Luo et al., 2022). Due to the high volatility in
our model simulations, the exponential nature of the implemented lake expansion and the nature of our merging algorithm, the range of individual growth rates is about four orders of magnitude higher than the observed ranges of 0.02 to 6.01 m/a (Jones et al., 2020) and 0.02 to 1.86 m/a (Luo et al., 2022), and should not be compared.

    The simulated water areas in our observation-based experiments increase over the simulation period of 100 years to just over 50%, which is a relatively high value that is very rare in current Arctic landscapes. While experiments for roughly the same
region in van Huissteden et al. (2011) also showed an initial increase of thawed area from 8% to about 25% in the first 70 years of their simulations, this initial increase is followed by a decrease in thawed area with some ensemble members reaching thawed areas as low as 2.2 and 2.7% after 100 years. The different behaviour of the two models can likely be explained with the different approaches regarding lake drainage. Our model takes all main parameter including drainage rates from remote sensing data directly, whereas van Huissteden et al. (2011) rely on an initialization with a river network. Even though our approach is
theoretically able to capture different drainage processes besides drainage through rivers, it still seems to underestimate lake drainage when the model is parametrized using current 20 year long remote sensing datasets.

    While our model does not require a-priori knowledge of sub-surface conditions, such as the model by van Huissteden et al. (2011), it is still very dependent on parametrization data that gives information on long-term developments of lake numbers and surface water areas within a landscape. Even though high-resolution remote sensing data with resolutions of less than 30 m
has been collected globally for more than two decades, the currently available data products (Schlaffer et al., 2012; Pekel et al., 2016; Nitze et al., 2017; Muster et al., 2017; Chen et al., 2023; Nitze and Nicholson, 2025) still do not provide the necessary consistency and coverage in both space and time to detect a robust trend in lake dynamics and to parametrize an influence of a climate forcing. The volatility of lake dynamics was shown to dominate a climate change signal in the 20 year dataset that we used. Additionally, issues with the remote sensing apparatus or the transmission can cause artefacts and data loss in the satellite
imagery, and lead to large parts of the data not being useable for our parameterization method, thereby making our parameter estimates less reliable. Timeseries of relative lake area changes that were derived from lake area dataset for our study region show a low signal-to-noise ratio of 0.37 and a volatility estimate that is 5.65 times higher than the drift estimate. Furthermore, gaps and artefacts in the satellite images that were used to derive the dataset, lead to 33% of data points that cannot be used for model parametrization. Because the model parameters are assumed to be region-specific, the parametrization should only
be done for domains that are relatively homogeneous in terms of geocryological and landscape conditions. This is why it is not always advisable to increase the size of parametrization datasets by including a larger area. Using a space-for-time approach for parametrization should also be handled with care for that reason. An increase of the length of the dataset would be the best option. A power analysis, with power of 0.8, significance level of 0.05, and the calculated signal-to-noise ratio as the effect size, suggests that a time series length of at least 60 years would be necessary to discern a signal in the investigated 40x40 km
cell. For the complete study region a signal-to-noise ratio of 0.40 and a minimum required timeseries length of 51 years was found. For this analysis, we used the python package *statsmodel* to solve the function for the power of a t-test for the minimum



sample size. Whether a climate influence could be quantified with such time series is still unclear. Even though high-resolution historical aerial imagery exists from the 1940s onwards for some regions across the Arctic (e.g. Muster et al.,2017), the gaps in the timeseries would likely still cause problems for a robust signal detection. A more extensive statistical analysis using

different synthetic datasets created by the model and our parametrization approach to reconstruct the input parameters, could help in formulating and quantifying requirements for parametrization datasets.

While the dependence on climate variables might become quantifiable for parameter $\mu$ and $\sigma$ in a few decades with continued retrieval of remote sensing data, the same is likely not true for $\lambda_f$ and $\lambda_d$. A robust estimation of these two parameters needs a number of observed formation and abrupt drainage events that satellite data will not be able to provide in the near future.

It might therefore be necessary to think about calibration approaches that are not solely depended on this type of data. Lake age databases could be used to create a paleo-record of lake formation events. Data on lake ages are sporadic, however, since determining lake ages usually involves the dating of sediment cores (e.g. Lenz et al., 2016), which are costly to obtain. Still, a synthesis study yielded a dataset of basal ages for 1207 lake across the high northern latitudes (Brosius et al., 2018, 2021, 2023). Anthony et al. (2014) also conducted radiocarbon dating methods on samples from 49 drained lake basins in the North Siberian

yedoma region. Recent advancement in detecting drained lake basins from satellite data (Wang et al., 2012; Bergstedt et al., 2021) could provide information on present-day drained area fractions in different regions and potentially be used to calibrate the model, especially in comparison with estimates of the timing of the drainage events. Research and data synthesis on the sub-surface capillary network of the Arctic tundra could further increase our understanding of drainage probability (Liljedahl et al., 2024).

## 5   Conclusions

This study provides a stochastic modelling attempt of regional changes in lake distributions and water area fractions in thermokarst-affected permafrost landscapes. The conceptual model accounts for processes of formation, expansion, merging, and drainage of lakes. Since thermokarst lake dynamics are highly dependent on local environmental conditions, their behaviour can be volatile on the landscape scale. With our suggested method, it is possible to parametrize this volatility and

thereby consider the differences in individual lake dynamics, without having to resolve small-scale landscape heterogeneities, which would not only require high-resolution modelling, but also extensive surface and sub-surface data, that is not available at the necessary scale.

We developed two variants of the conceptual model. In a first variant, new lakes are implicitly not allowed to form on previously drained area, while in the second variant lakes can expand onto previously drained area and also form on it. Both

variants deliver similar results for simulations for our coastal study site in the Yana-Indigirka Lowland and over a timescale of 100 years. To the best of our knowledge, our model is the first to consider the interactions between active and drained lakes as represented in the variants. While both variants are highly simplified representations, they constitute an important first step towards a more complex and realistic model, and give an opportunity to investigate the impacts of the two underlying assumptions on lake dynamics as well as the relative importance of the involved processes.



We tested three different regimes of the model dynamics: complete drainage, oscillation, and stabilization of the lake area. All these regimes are possible and could be relevant for particular geographical regions. For example, the oscillation regime reflects the long-term development of a lake-rich area with periodic establishment and draining.

We attempted to quantify model parameters using existing remote sensing data. However, it is difficult to validate simulation results with the current state of observational data. Even though the model is conceptual, it is still very dependent on the 650 parameterization data. Currently available remote sensing based time series of surface water are at maximum 40 years long, but mostly shorter and have many data gaps, particularly in the high latitudes. Longer term data (centuries to millennia) are very scarce and less precise as they are typically based on soil samples and geochronological dating. Furthermore, thermokarst lake dynamics are volatile, making it hard to detect a climate influence in the available datasets using our method. It is therefore not yet possible to accurately parametrize or calibrate the lake dynamics response to climate forcing, and use the model for 655 projections under different climate change scenarios. This currently limits the potential of application of the model for a large-scale ESM.

The parameterization of lake expansion and gradual drainage might become possible with slightly longer time series. A first power analysis suggests that at least 50 years of observational data would be necessary to detect a signal of changes in lake area dynamics within the study region. However, the lack of data on longer time scales will still prevail and formation and abrupt 660 drainage rate likely can't be captured with a few more decades of satellite data retrieval. It is therefore necessary to think about different calibration approaches for these two parameters.

*Code and data availability.* The model code will be made publicly available via Zenodo with the publication of the final version of this manuscript. It can currently be found under https://github.com/cowniflow/TLM. The dataset of lake areas used as parameterization data is available on Zenodo under https://doi.org/10.5281/zenodo.15011121.





**Appendix A: Modelling Scheme**

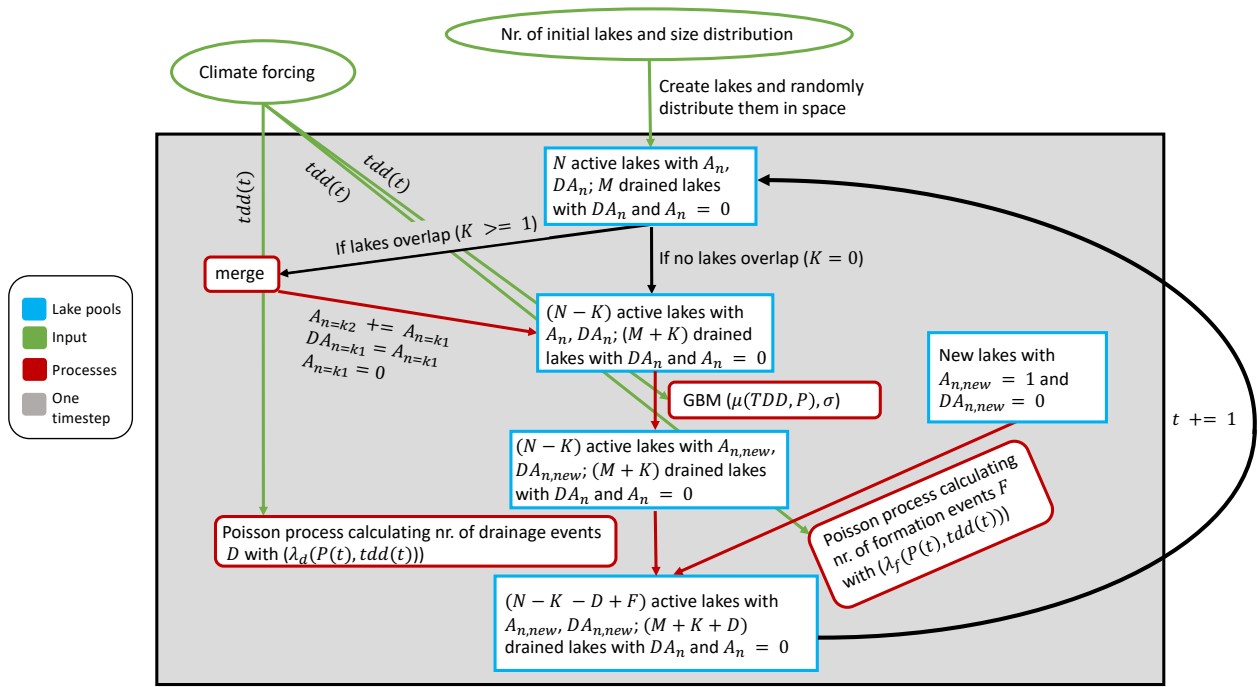

**Figure A1.** Modelling scheme. Green indicates input into the model, i.e. initialization data and a climate forcing. The climate forcing consists of an annual time series for precipitation $P$ and thaw degree days (TDD). The grey box contains all operations that happen at each time step. Blue indicates the model objects, i.e. the active and inactive lake pool. Red indicates the main computations and processes, which are: merging of overlapping lakes, geometric Brownian motion (GBM) simulating lake expansion and gradual drainage, a Poisson process that calculates the number of drainage events, and a Poisson process that calculates the number of formation events, i.e. new lakes that are added to the lake pool. $N$ is the initial number of active lakes at each timestep, $M$ is the initial number of inactive lakes. $A_n$ is the water surface area of the $n$-th lake, whereas $DA_n$ is its drained area. $A_{n,new}$ and $DA_{n,new}$ are both of these areas after applying expansion or gradual drainage via GBM. $K$ is the number of overlaps and therefore merging events. $D$ is the number of drainage events, and $F$ the number of formation events. $P(t)$ and $TDD(t)$ are the values of the climate variables precipitation and thaw degree days at time $t$.

.



## Appendix B: Climate Correlation Analysis

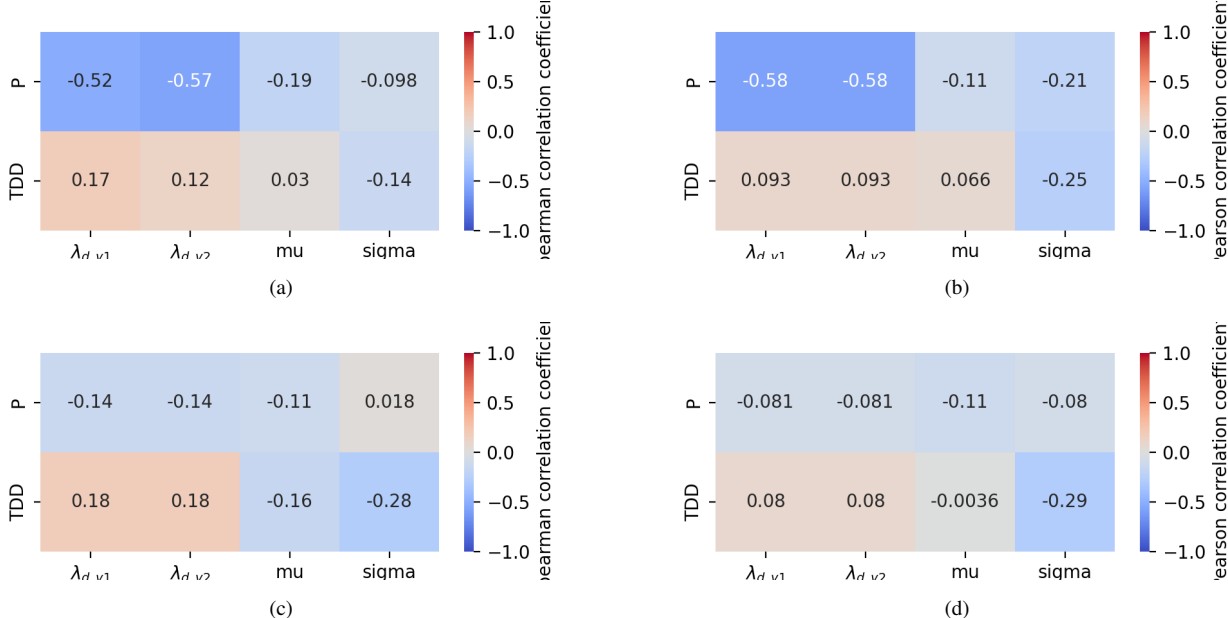

**Figure B1.** Spearman (a) and Pearson (b) correlation coefficient between parameters and that year's climate variables thawing degree days (TDD) and annual precipitation P in whole the study area region between 2000 and 2020, as well as Spearman (c) and Pearson (d) correlation coefficient between parameters and that year's climate variables TDD and annual precipitation P in the 40x40 km cell.

*Author contributions.* Constanze Reinken developed the model and parameterization method, wrote the model code, executed the simulations and analysis, and drafted the manuscript with input and corrections from all other authors. Victor Brovkin, Philipp de Vrese, and Constanze Reinken collaborated on the research question, project approach, and overall storyline of the manuscript. Ingmar Nitze provided

the remote sensing data product used in the study. Helena Bergstedt, Guido Grosse and Ingmar Nitze offered insight into the state of remote sensing data and data products in the Arctic as well as dynamics of thermokarst lakes and lake drainage.

*Competing interests.* Philipp de Vrese is a member of the editorial board of The Cryosphere.

*Acknowledgements.* This work was supported by the European Research Council project Q-Arctic (grant no. 951288). It used resources of the Deutsches Klimarechenzentrum (DKRZ) granted by its Scientific Steering Committee (WLA) under project ID bm1236. Further, datasets

provided by Tobias Stacke (Max-Planck Institute for Meteorology) via the DKRZ data pool were used. We want to thank Annett Bartsch and Clemens von Baeckmann from b.geos for their support in navigating remote sensing data publications.



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
