# Peer review of "Stochastic Modelling of Thermokarst Lakes: Size Distributions and Dynamic Regimes"

_EGUsphere, 2025_

## Referee Comment (RC2)

**Referee Report**

The manuscript develops a stylised stochastic framework for thermokarst–lake dynamics at landscape scale. Formation and abrupt drainage are represented as Poisson processes with intensities scaled by regional area fractions (two variants). Individual lake areas evolve via Geometric Brownian Motion (GBM). An area-fraction cap can limit further growth. Parameters are inferred from annual lake-area time series; three qualitative regimes (complete drainage, oscillation, quasi-stabilisation) are demonstrated. The modelling separation between birth/death (Poisson) and multiplicative size dynamics is mathematically attractive and invites analysis and principled calibration.

**Major comments**

(1) There may be an issue with the GBM discretisation and Brownian scaling. Please implement the standard GBM increment for time step $\Delta t$:

$$\log \frac{a_i(t)}{a_i(t - \Delta t)} = (\mu - \tfrac{1}{2}\sigma^2)\, \Delta t + \sigma \sqrt{\Delta t}\, Z, \quad Z \sim \mathcal{N}(0, 1).$$

The text describes sampling the Normal with standard deviation equal to $dt$ (the time step). This should be $\sqrt{dt}$. If steps remain annual, state $\Delta t = 1$ year explicitly and correct the noise scaling.

(2) Linear area-fraction scaling implies constant hazards per unit area and no explicit dependence on lake size, clustering, or covariates. A modest generalisation with state-dependent intensities $\lambda_f(\cdot), \lambda_d(\cdot)$ (e.g. log-link GLM or Cox hazards) would capture simple nonlinear feedbacks while remaining identifiable.

(3) Specify the boundary condition at zero area (absorbing vs. truncation) and whether very small areas are killed. This choice affects the conceptual split between gradual (GBM) and abrupt (Poisson) drainage under annual sampling.

(4) Beyond merging, lakes evolve independently. Introducing weak correlation (shared random environment / spatial frailties) or lightly correlated GBM shocks could capture hydrologic connectivity without materially increasing complexity.

(5) The current merging rule can produce implausibly large single lakes and is computationally heavy. A stochastic, geometry-consistent alternative (continuum percolation / Boolean union-of-sets with polydisperse footprints), optionally with post-merge fission, would improve realism and cluster statistics.

(6) The birth–growth–death structure invites analysis:

- a size-structured Fokker–Planck (McKendrick–von Foerster + diffusion) for the area density $p(a, t)$ with integral birth/death terms;
- conditions for stationarity or self-similarity (lognormal-type tails) and closed-form moment dynamics;

- expectation dynamics for the total water fraction under an $A_{\text{lim}}$ ceiling.

Even partial moment equations would substantively strengthen the mathematical core.

**(7)** With annual data and gaps, the drift $\mu$ is weakly identified relative to the volatility $\sigma$. Consider a composite-likelihood or state-space formulation with uncertainty bands; Bayesian pooling (across neighbouring cells) can stabilise $\mu$ and the hazards.

**(8)** Simple contemporaneous or one-year-lag correlations have low power and can miss nonlinearity. Try distributed-lag specifications, information-criteria-based lag selection, partial correlations (conditioning on antecedent water fraction), or spline thresholds before concluding there is no relationship.

**(9)** State the units of $\lambda_f, \lambda_d$ (e.g. events per year per area) and specify how rates scale under aggregation/disaggregation so ESM tiles can ingest them consistently.

**(10)** Abrupt drainage = (near) complete loss within a year via the Poisson process; gradual drainage = negative GBM drift. With annual sampling, large partial losses (e.g. 60–90%) can be ambiguous; a competing-risks view with size-dependent abrupt-drainage hazard would help.

**(11)** Provide a short $\Delta t$ sensitivity (e.g. semiannual with rescaled noise) to test robustness of annual stepping.

**(12)** Complement the qualitative regimes with simple statistics (cluster count, Gini coefficient of lake areas, distribution quantiles) that distinguish regimes numerically.

**Minor comments**

**(13)** Bring the Variant 1/2 definitions forward and summarise implications in a small table. Clarify whether $A_{\text{lim}}$ is a hard cap (projection) or a soft ceiling (e.g. logistic drift modulation).

**(14)** Clearly state the initial area assigned to newly formed lakes.

**(15)** Collect symbols/units in one table (including whether $\mu, \sigma$ are per-year) to aid reproducibility.

**Overall recommendation**

The framework is promising for ESM-facing parameterisation. I recommend: (i) correcting the GBM noise scaling, (ii) formalising state-dependent event rates, (iii) adding basic analytical results (moments / Fokker–Planck), and (iv) replacing or augmenting the merging rule with a stochastic, geometrically consistent alternative. These steps would materially strengthen the rigour of the developed approach.

---

## Author Comment (AC2)

Point-by-point replies to each comment are listed below.

1. **Comment:** *The application of Geometric Brownian Motion (GBM) and Poisson processes to model lake area change and formation/drainage events is innovative and well-justified within the context of landscape-scale heterogeneity and stochasticity. This is a significant step beyond previous deterministic or analytical models.*

   **Reply:** Thank you for this comment, which emphasizes the innovative nature of our modelling approach.

2. **Comment:** *The three idealized regimes (Complete Drainage, Oscillation, Stabilization) effectively demonstrate the model's behavioral range and provide a useful conceptual framework for understanding possible long-term trajectories of thermokarst landscapes. The links to real-world examples for each regime are well-made.*

   **Reply:** Thank you for this comment, which mentions that our presentation of the three idealized regimes provides an effective demonstration of the model's behavioural range and contains well-made links to real-world examples.

3. **Comment:** *The current merging algorithm is identified correctly as a significant weakness. The assumption that merged lake area is the sum of the two original areas and that the new lake is perfectly circular is physically unrealistic and computationally expensive. It likely leads to a drastic overestimation of lake sizes and an underestimation of lake numbers.*

   **Reply:** We agree that the merging algorithm is a significant weakness. We are planning to test a new approach following suggestions from Reviewer 2 as part of the revision process. However, we argue that even without this change, a publication could still be beneficial for the permafrost modelling community and invite further model development led by other researchers with more expertise in potentially useful fields such as percolation theory. As mentioned in the comment, the weakness of the current merging algorithm is already addressed in the current version of the manuscript.

4. **Comment:** *The high percentage of unusable data points (30-33%) in the remote sensing dataset severely impacts the robustness of the parameter estimation. This uncertainty propagates through the observation-based simulations and limits the confidence in the derived parameters ($\lambda f$, $\lambda d$, $\mu$, $\sigma$).*

   **Reply:** Thank you for highlighting the issue of low confidence in parameter estimates due to gaps in the observational data.

5. **Comment:** *A crucial result is that the stochastic component ($\sigma$) dominates the deterministic drift ($\mu$) in the observed period. This implies that, for the 2000-2020 period in this region, random environmental variability was a stronger driver of annual lake area changes than any clear climate-driven trend, explaining the lack of strong correlations.*

**Reply:** Thank you for highlighting this. We will emphasize this point more in the "Conclusions". Furthermore, we will review this finding after testing the methods described in comments 7 and 8 by Reviewer 2.

6. **Comment:** *The inability to find significant correlations between model parameters and climate variables (TDD, P) is a notable negative result. While honestly reported, it highlights the current impossibility of confidently projecting lake dynamics under climate change scenarios with this model, as intended in the abstract. This is a major constraint on its immediate application in ESMs.*

   **Reply:** We agree that this issue significantly limits the immediate application of the model in ESMs. We will address implications for ESMs more directly in the "Discussion" and "Conclusions". Before that, however, we will review this finding following comment 8 by Reviewer 2.

7. **Comment:** *The observation-based simulations project water area fractions increasing to over 50%, which is acknowledged as rare. This, combined with the high volatility, suggests the model parameters derived from 20 years of data may not be stable or representative of centennial-scale dynamics, potentially overestimating expansion.*

   **Reply:** We agree that the results of the observation-based simulations suggest that the model parameter derived from 20 years of data might not be suitable for centennial-scale dynamics. We will add one or two sentences to the "Discussion" that mentioning this directly.

8. **Comment:** *The suggestion that the idealized simulations could be interpreted as spanning 10 ka with a 10-year time step is helpful for context, but the parameters would then be "per decade." This should be stated explicitly in the text to avoid confusion (e.g., in Table 1, add a note: "Parameters are per year; for a 10-year time step interpretation, values would be per decade").*

   **Reply:** Thank you for the suggestion of adding a note to Table 1 regarding the interpretability of hypothetical parameter values as being "per decade", and will implement this.

9. **Comment:** *The comparison with the van Huissteden et al. (2011) model is good. The explanation for the differing results (their reliance on a prescribed river network vs. your data-driven drainage rates) is plausible and highlights the advantage of your approach, but also its current data dependency.*

   **Reply:** Thank you for this comment, which mentions the usefulness of our comparison with the model by van Huissteden et al. 2011.

10. **Comment:** *The definition of "abrupt drainage" as a complete (>90% loss) and rapid event is clear. However, the discussion of results from other studies (Jones et al., 2011, 2020) that use different thresholds (e.g., >25% loss) is slightly confusing. A*

*small table summarizing different study's definitions and converting their rates to a common framework would be helpful.*

**Reply:** We agree that the current comparison with other studies (Jones et al. 2020 and Jones et al. 2011) on lake drainage is confusing considering the different thresholds that were used to define drainage events. To improve the comparison between these two studies with ours, we are planning to identify all lakes in our simulations that have lost at least 25% of their area. Since there are no standard thresholds or definitions for different drainage types, as far as we are aware, this number is somewhat arbitrary, but in line with the definition in Jones et al. 2011 & 2020. This will effectively also include our definition of gradual drainage and make a comparison more justified. With this change, we do not see a benefit of a table, as we will only use one definition of lake drainage for the comparison.

11. **Comment:** *The axis titles of many figures are obscured, or the figures are not very clear, for example, in Figures 5, 6 and B1.*

    **Reply:** Thank you for pointing this out. We will revise the figures to include fully readable axis and axis titles.

12. **Comment:** *There are some spelling errors in the manuscript. For instance, the first reference should read "thermokarst lake" instead. The authors should carefully proofread the manuscript to avoid such errors.*

    **Reply:** Thank you for pointing this out. We will carefully proof-read the manuscript.

---

## Author Comment (AC3)

Point-by-point replies to each comment are listed below.

Major comments:

1. **Comment:** *There may be an issue with the GBM discretisation and Brownian scaling. Please implement the standard GBM increment for time step Δt. The text describes sampling the Normal with standard deviation equal to dt (the time step). This should be √dt. If steps remain annual, state Δt= 1 year explicitly and correct the noise scaling.*

   **Reply:** We sincerely thank you for pointing out this potential error. This was a mistake in the description only and done correctly in the computations. We will correct the text accordingly and state the yearly time step more explicitly.

2. **Comment:** *Linear area-fraction scaling implies constant hazards per unit area and no explicit dependence on lake size, clustering, or covariates. A modest generalisation with state-dependent intensities λf (·),λd(·) (e.g. log-link GLM or Cox hazards) would capture simple nonlinear feedbacks while remaining identifiable.*

   **Reply:** We agree that the linear area-fraction scaling is a simplification that implies constant hazards unrelated to lake size, clustering or covariates and thank the reviewer for their suggestion to generalize this. However, we believe that the influence of these factors on formation and drainage probability is difficult to measure. In light of that, we argue that the current linear scaling is a sufficient simplification and that a generalization with a log-link GLM or the Cox hazards method without these quantifications will not be insightful. We are going to give further thought to how these methods could be implemented without adding additional degrees of freedom to the model in our forthcoming research beyond this manuscript.

3. **Comment:** *Specify the boundary condition at zero area (absorbing vs. truncation) and whether very small areas are killed. This choice affects the conceptual split between gradual (GBM) and abrupt (Poisson) drainage under annual sampling.*

   **Reply:** We will add a clearer explanation of the treatment of gradually vs. abruptly drained lakes in the model:

   - Abruptly drained lakes are removed from the pool of active lakes and will therefore stay drained throughout the rest of the simulation. This means that abruptly drained lakes are effectively killed.
   - Gradually drained lakes are not removed from the pool of active lakes. Small areas as a result of gradual drainage are therefore not killed, but can be subject to expansion or further gradual drainage. However, if an area has reached a value of zero due to gradual drainage, it will effectively stay zero for the rest of the simulation due to the nature of the SDE (GBM) that is used to model lake area evolution.

4. **Comment:** *Beyond merging, lakes evolve independently. Introducing weak correlation (shared random environment / spatial frailties) or lightly correlated GBM shocks could capture hydrologic connectivity without materially increasing complexity.*

   **Reply:** While we agree that weak correlation in lake area evolution would not materially increase complexity in terms of computation, we do want to emphasize that it would add another degree of freedom to the model, i.e. another parameter that would need to be calibrated using observational data. Considering that we found parameterization / calibration of the model to be difficult with currently available remote sensing datasets, we argue that it could be more beneficial to keep this first model version simple and conceptual with as few parameters as possible. Following the argumentation by Victorov et al. 2019, we believe that independent lake evolution is a justified assumption for a first modelling approach. We are happy to give further thought to this, but do not consider it a priority at this point.

5. **Comment:** *The current merging rule can produce implausibly large single lakes and is computaionally heavy. A stochastic, geometry-consistent alternative (continuum percolation / Boolean union-of-sets with polydisperse footprints), optionally with post-merge fission, would improve realism and cluster statistics.*

   **Reply:** We strongly agree that a stochastic and geometry-consistent alternative for lake merging could significantly improve realism of model simulations as well as computational requirements. We sincerely thank you for your input on this issue. We will try to implement a Boolean union-of-sets approach based on clustering overlapping lakes rather than merging them. This way we could still keep track of the individual circles / disks and potentially allow them to separate again (post-merge fission). Such an approach can also lead to more realistic representation of the surface area of merged lakes and their shape. While we still need to get an overview on the details of such an approach and possibly useful functions or packages, we think that an implementation is feasible.

6. **Comment:**
   *The birth–growth–death structure invites analysis:*
   > *• a size-structured Fokker–Planck (McKendrick–von Foerster + diffusion) for the area density p(a,t) with integral birth/death terms;*
   > *• conditions for stationarity or self-similarity (lognormal-type tails) and closed-form moment dynamics;*
   > *• expectation dynamics for the total water fraction under an $A_{lim}$ ceiling.*
   *Even partial moment equations would substantively strengthen the mathematical core.*

   **Reply:** Thank you for your suggestions on additional analysis of the model dynamics. We think that a stronger focus on changes in the PDF of lake areas under different regimes and parameterizations makes sense for the paper, especially after improving the implementation of lake merging. We therefore agree that a Fokker-Planck equation that describes the time evolution of the PDF or partial moment equations of lake numbers, average lake area and area variance would be a useful addition to the analysis of the model output, and will include this. Depending on our available

time, we will also consider analyzing under which conditions the PDF becomes stationary or self-similar and how $A_{lim}$ influences expected behavior of total water fraction, but we do not see this as a priority at this point.

7. **Comment:** *With annual data and gaps, the drift μ is weakly identified relative to the volatility σ. Consider a composite-likelihood or state-space formulation with uncertainty bands; Bayesian pooling (across neighbouring cells) can stabilise μ and the hazards.*

   **Reply:** Thank you for your suggestions on how to improve identification of the drift parameter in observational data, which is otherwise difficult to detect due to high volatility and data gaps. We note that we already excluded data points (i.e. lake area for an individual lake in a specific year) from the calculation if they contained measurement gaps (i.e. an area within a lake polygon that was categorized as "no data" due to gaps in the satellite data). At this stage, we are not sure if the suggested methods would lead to an improvement, but will consider and potentially test this.

8. **Comment:** *Simple contemporaneous or one-year-lag correlations have low power and can miss nonlinearity. Try distributed-lag specifications, information-criteria-based lag selection, partial correlations (conditioning on antecedent water fraction), or spline thresholds before concluding there is no relationship.*

   **Reply:** Thank you for these suggestions. We understand that the current investigation of a correlation between climate and our lake dynamics parameters is too simplified and appreciate your input on this. We are happy to test the mentioned methods and are prepared to adjust our conclusions regarding a climate influence on lake dynamics according to the new results.

9. **Comment:** *State the units of λf ,λd (e.g. events per year per area) and specify how rates scale under aggregation/disaggregation so ESM tiles can ingest them consistently.*

   **Reply:** Thank you for pointing this out. We will make sure that the unit (event per year per $m^2$) for our parameters for formation and abrupt drainage is stated clearly everywhere in the paper. The rates should be multiplied by the time step and the relevant area (depending on model variant) within one or several ESM tiles. In the case of several tiles, the multiplication needs to be done with the sum of the relevant area across these tiles. The relevant areas are calculated according to eq. 9 or 10 depending on the model variant. In addition to these equations, we will also add equations for the scaling of the parameters directly.

10. **Comment:** *Abrupt drainage = (near) complete loss within a year via the Poisson process; gradual drainage = negative GBM drift. With annual sampling, large partial losses (e.g. 60–90%) can be ambiguous; a competing-risks view with size-dependent abrupt-drainage hazard would help.*

    **Reply:** We interpret this comment to refer to the analysis of / parameterization from observational data rather than model development or analysis of model results, since

the model itself can keep track of whether a lake was abruptly or gradually drained. However, large partial losses of lake area in observational data can indeed be ambiguous, making it difficult to estimate an abrupt drainage rate. We therefore sincerely appreciate this suggestion. However, we think that determining such a size-dependent abrupt-drainage hazard accurately enough to significantly improve estimates of abrupt drainage rate from observational data, would go beyond the scope of this paper. Instead of incorporating this into our method, we will add this point to the "Discussion" as an outlook for improving parameterization approaches.

11. **Comment:** *Provide a short Δt sensitivity (e.g. semiannual with rescaled noise) to test robustness of annual stepping.*

    **Reply:** We see how a short time step sensitivity study can be beneficial to test robustness and will conduct one. We also want to note, however, that our model does not contain explicit seasonal dynamics and that we therefore do not recommend using it with a time step of less than a year.

12. **Comment:** *Complement the qualitative regimes with simple statistics (cluster count, Gini coefficient of lake areas, distribution quantiles) that distinguish regimes numerically.*

    **Reply:** Thank you for this suggestion. However, we are not yet sure how much additional information they can provide to the Fokker-Planck or partial moment equations. We will need to look more into the suggested measures.

Minor comments:

13. **Comment:** *Bring the Variant 1/2 definitions forward and summarise implications in a small table. Clarify whether $A_{lim}$ is a hard cap (projection) or a soft ceiling (e.g. logistic drift modulation).*

    **Reply:** We agree that the definitions of Variant 1 and 2 should be put more into the foreground and made clearer, e.g. through a table - as suggested by the reviewer. Besides adding such a table, we will also add further equations that directly show the calculation of the formation and drainage probability (also see comment 9), which will aid in clarifying the differences in the two variants.
    The area fraction limit $A_{lim}$ is a hard cap for lake expansion. When it is reached, no expansion of existing surface areas is allowed and lakes can therefore only shrink at that point, until the area fraction is below the limit again. Formation, however, is not explicitly inhibited. Instead, the formation probability will typically be zero at $A_{lim}$ due to the implementation of the area-fraction scaling. We will make this clearer in the text.

14. **Comment:** *Clearly state the initial area assigned to newly formed lakes.*

    **Reply:** In the currently included simulations, the initial area of lakes was 1 km$^2$. We will make sure to state this more clearly. After upgrading parts of the model, we will

redo the simulations and adjust the initial to 1 ha to be in line with the resolution of the observational data.

15. **Comment:** *Collect symbols/units in one table (including whether μ,σ are per-year) to aid reproducibility.*

    **Reply:** We thank the reviewer for suggesting this and agree that this would significantly increase clarity. We will include this in a revised version of the manuscript.